# Dynamics of microcompartment formation at the mitosis-to-G1 transition

Viraat Y. Goel [1,2,3,4], Nicholas G. Aboreden[5,6], James M. Jusuf[1,2,3,4], Haoyue Zhang [7], Luisa P. Mori[6], Leonid A. Mirny [8], Gerd A. Blobel [5,6], Edward J. Banigan [8]✉ & Anders S. Hansen [1,2,3,4]✉

As cells exit mitosis and enter G1, chromosomes decompact and transcription is reestablished. Hi-C studies have indicated that all interphase three-dimensional genome features, including A/B compartments, topologically associating domains and CCCTC-binding factor loops, are lost during mitosis. However, Hi-C is insensitive to features such as microcompartments, nested focal interactions between cis-regulatory elements. Here we apply region capture Micro-C to mouse erythroblasts from mitosis to G1. We unexpectedly observe microcompartments in prometaphase, which strengthen in anaphase and telophase before weakening throughout G1. Microcompartment anchors coincide with transcriptionally spiking promoters during mitosis. Loss of condensin loop extrusion differentially impacts microcompartments and A/B compartments, suggesting that they are partially distinct. Polymer modeling shows that microcompartment formation is favored by chromatin compaction and disfavored by loop extrusion, providing a basis for strong microcompartmentalization in anaphase and telophase. Our results suggest that compaction and homotypic affinity drive microcompartment formation, which may explain transient transcriptional spiking at mitotic exit.

The three-dimensional (3D) structure and function of the genome are linked throughout the cell cycle as chromatin reorganizes to facilitate cell growth and division. During mitosis, chromosomes change in both structure and function and, as the nuclear envelope breaks down, chromosomes compact -1.5–3-fold, CCCTC-binding factor (CTCF), cohesin and many transcription factors are evicted and transcription is largely shut off[1–15]. These changes include the loss of all Hi-C-observable 3D genome structures evident in interphase, including A/B compartments, topologically associating domains (TADs), structural CTCF/cohesin loops and functional loops between cis-regulatory elements (CREs), such as enhancers and promoters, by the time cells reach prometaphase (PM)[1,14–25]. This has led to the paradigm that interphase 3D genome structure is entirely lost in mitosis and must be rebuilt de novo as cells enter G1. Indeed, Hi-C studies have demonstrated that, starting in anaphase and telophase (AT), A/B compartments, TADs and CTCF/cohesin loops form slowly and gradually strengthen to reach full strength by late G1 (LG1)[16–21,25] and some CRE loops begin to appear by AT[1,17,18,22,26]. However, most CRE loops are poorly resolved by Hi-C and regulated differently than CTCF/cohesin-mediated structural loops[27,28], which led us to revisit the paradigm that all interphase 3D features are entirely lost in PM.

[1]Department of Biological Engineering, Massachusetts Institute of Technology, Cambridge, MA, USA. [2]Gene Regulation Observatory, Broad Institute of MIT and Harvard, Cambridge, MA, USA. [3]The Novo Nordisk Foundation Center for Genomic Mechanisms of Disease, Broad Institute of MIT and Harvard, Cambridge, MA, USA. [4]Koch Institute for Integrative Cancer Research, Cambridge, MA, USA. [5]Perelman School of Medicine, University of Pennsylvania, Philadelphia, PA, USA. [6]Division of Hematology, The Children's Hospital of Philadelphia, Philadelphia, PA, USA. [7]Institute of Molecular Physiology, Shenzhen Bay Laboratory, Shenzhen, China. [8]Institute for Medical Engineering and Science and Department of Physics, Massachusetts Institute of Technology, Cambridge, MA, USA. ✉e-mail: ebanigan@mit.edu; ashansen@mit.edu

To overcome the detection limits of Hi-C, we recently developed region capture Micro-C (RCMC)[28]. RCMC combines Micro-C, which is uniquely sensitive to CRE loops[27,29–31], with a tiling capture step to concentrate sequencing reads in regions of interest (ROIs)[28,32]. This allows RCMC to achieve ~100–1,000-fold higher data depth in target regions than possible with genome-wide Hi-C or Micro-C for a comparable number of sequencing reads. Using RCMC, we discovered highly nested focal interactions between CREs that were previously undetectable[28]. We termed these structures 'microcompartments' because they are largely robust to loss of cohesin-based loop extrusion and appear to form through an affinity-mediated compartmentalization mechanism akin to block copolymer microphase separation[28,33,34]. Accordingly, microcompartments denote both a 'grid of dots' pattern in contact maps (representing nested focal interactions) and a mechanism of interaction (affinity-mediated compartmentalization). Microcompartmental 'dots' result from loops largely formed between CRE anchors.

Given that all 3D genome structural features were shown by Hi-C to be lost in mitosis, we chose this system to explore the mechanisms and dynamics of microcompartment formation. We applied RCMC to mouse erythroid cells across the mitosis-to-G1 (M-to-G1) transition. Unexpectedly, we observe microcompartments in mitosis, in contrast to all prior Hi-C studies reporting that chromosomes lose all 3D genome structural patterns during cell division[1,14–25]. Furthermore, we find that microcompartments transiently peak in strength in AT before gradually weakening in G1 and microcompartment dynamics correlate with transient transcriptional spiking at the M-to-G1 transition. Integrating 3D polymer modeling, we show how an interplay of affinity, extrusion activity and chromosome compaction can explain these structural dynamics. This provides a mechanistic framework for understanding how loop extrusion, compaction and affinity-mediated compartmentalization govern 3D genome folding across structural scales and across the cell cycle.

## Results

### RCMC resolves 3D genome folding dynamics from mitosis to G1

To resolve ultrafine-scale 3D genome folding dynamics following mitosis, we used the experimental system previously established and validated by Zhang et al.[18]. We purified synchronized mouse G1E-ER4 erythroblasts using fluorescence-activated cell sorting (FACS) on the basis of the signal from mCherry fused to the cyclin B mitotic degradation domain (mCherry–MD) and the DNA content to achieve ~98% pure PM, AT, early G1 (EG1), mid-G1 (MG1) and LG1 cell populations (Fig. 1a and Extended Data Fig. 1). We then performed RCMC[28] to generate deep contact maps at five diverse regions selected for their density of CREs (Extended Data Figs. 1b and 2–7). Such maps allow us to sharply resolve and follow genomic structures across scales of organization through mitotic exit, including A/B compartments, TADs, loops and microcompartments, which are invisible in sparser datasets (Fig. 1a,b and Extended Data Figs. 2 and 4–8).

We obtained the expected interaction scaling with genomic distance, $P(s)$, for interphase and mitosis[16,18,19,25] and observed first-derivative peaks of ~400 kb in mitosis and ~50 kb in G1 (Fig. 1c), which correspond approximately to the average extruded loop sizes[35–38]. Comparing our RCMC maps to prior Hi-C data[18], we observed the same gradual strengthening of large A/B compartments and bottom-up formation of TADs and CTCF loops upon mitotic exit, thus validating the correspondence between RCMC and Hi-C at coarse resolution (Extended Data Figs. 3–7). Critically, our RCMC maps are between ~100-fold and ~1,000-fold deeper than the Hi-C data[18] (Fig. 1d and Extended Data Figs. 1c and 2b), are highly reproducible (Extended Data Fig. 1d,e) and have similar read coverage profiles as Hi-C (Extended Data Fig. 3). Data depth was confirmed by downsampling the RCMC data ~512–1,024-fold, which yielded comparable data depth (Extended Data Fig. 2c) and contact maps (Extended Data Fig. 2d) as

Hi-C[18]. Having validated the quality of our RCMC data, we next explored the dynamics of chromatin structure formation during mitotic exit.

### RCMC reveals nested focal looping interactions between CREs during mitosis

Visualizing our RCMC maps at fine resolutions revealed a dramatic restructuring of chromosomes across the cell cycle (Fig. 2a and Extended Data Figs. 4–7). Strikingly, the maps reveal strong loops across each locus in PM and AT. Thus, while we also observe an absence of TADs and A/B compartments in PM consistent with prior work, our result overturns the paradigm that all 3D genome structure is lost in PM[1,14–25]. To quantify these dynamics, we annotated dots in the contact maps, which correspond to focal interactions between two sites, hereafter referred to as 'loops'. The superset of loops formed across the M-to-G1 transition over the five RCMC regions (spanning ~7 Mb in total) yielded 3,350 loops formed by 363 anchors (Fig. 2a and Extended Data Figs. 4–7); by contrast, only 134 of the 3,350 loops resolved by RCMC were detectable in Hi-C data (Extended Data Fig. 8). If we scale this number of loops to the whole genome, assuming the number is proportional to the number of genes, we estimate ~346,000 loops across the genome, which is in line with recent estimates in both mouse[39] and human[40] cells. Annotated loops spanned all length scales within captured loci, with a mean length of 368 kb (Fig. 2b) and most loops were formed by a subset of high-connectivity anchors, with many dozens of anchors forming ~40–50 loops (Fig. 2c). Lastly, we tested whether loops could be artifacts of chromatin accessibility[41] at loop anchors (Supplementary Fig. 1); ~40% of all loop anchors did not coincide with DNase-Seq peaks[41] (Supplementary Fig. 1b) and CREs are less accessible in mitosis than in G1 (ref. 41), thus ruling out accessibility bias.

To classify loops by their functional identity, we intersected loop anchors with gene promoter annotations (transcription start sites (TSSs)), epigenetic markers of enhancers (H3K27ac and H3K4me1) and structural looping factors (CTCF and the cohesin subunit RAD21), which revealed most anchors to be promoters and enhancers (Fig. 2d, Supplementary Fig. 2 and Supplementary Table 1). Indeed, we found most loops to be CRE loops (Fig. 2e): only ~1% of loops (34/3,350) were 'structural loops' lacking CRE overlap, anchored solely by CTCF/cohesin on both sides. Instead, ~90% of all loops were CRE-anchored on one side and ~80% CRE-anchored on both sides (P–P, E–P or E–E, where P is promoter and E is enhancer). Revisiting the number of loops formed per anchor (Fig. 2c) revealed that promoters and enhancers comprise nearly all the highest-connectivity anchors, whereas CTCF/RAD21 anchors form far fewer loops (Fig. 2f). The high connectivity of CRE anchors contrasting with CTCF-anchored loops is consistent with a different interaction mechanism for CREs, such as affinity between similar chromatin states and/or transcription factors. Thus, CRE anchors form many loops leading to microcompartment formation (grid of dots pattern; Fig. 1b), unlike CTCF/cohesin anchors, which form few loops.

Notably, we observed that microcompartments strikingly change across the M-to-G1 transition (Fig. 2a and Extended Data Fig. 4–7). Microcompartments are visible in PM, before increasing in strength relative to background in AT and then gradually weakening upon G1 entry with many microcompartmental loops erased by LG1 (Fig. 2a and Extended Data Figs. 4–7). Quantitative analysis confirmed this observation; the CRE loops that make up microcompartments (P–P, E–P and E–E) peak in strength in AT (Fig. 2g and Supplementary Fig. 3a). To better characterize the unexpected transience of microcompartments, we next explored the strengths of different loop types across the M-to-G1 transition.

### Microcompartments transiently strengthen and then weaken across the M-to-G1 transition

To further explore the dynamics of microcompartmentalization, we began by visualizing representative examples of microcompartmental CRE loops (Fig. 3a, i–iv) and structural CTCF loops (Fig. 3a, v)

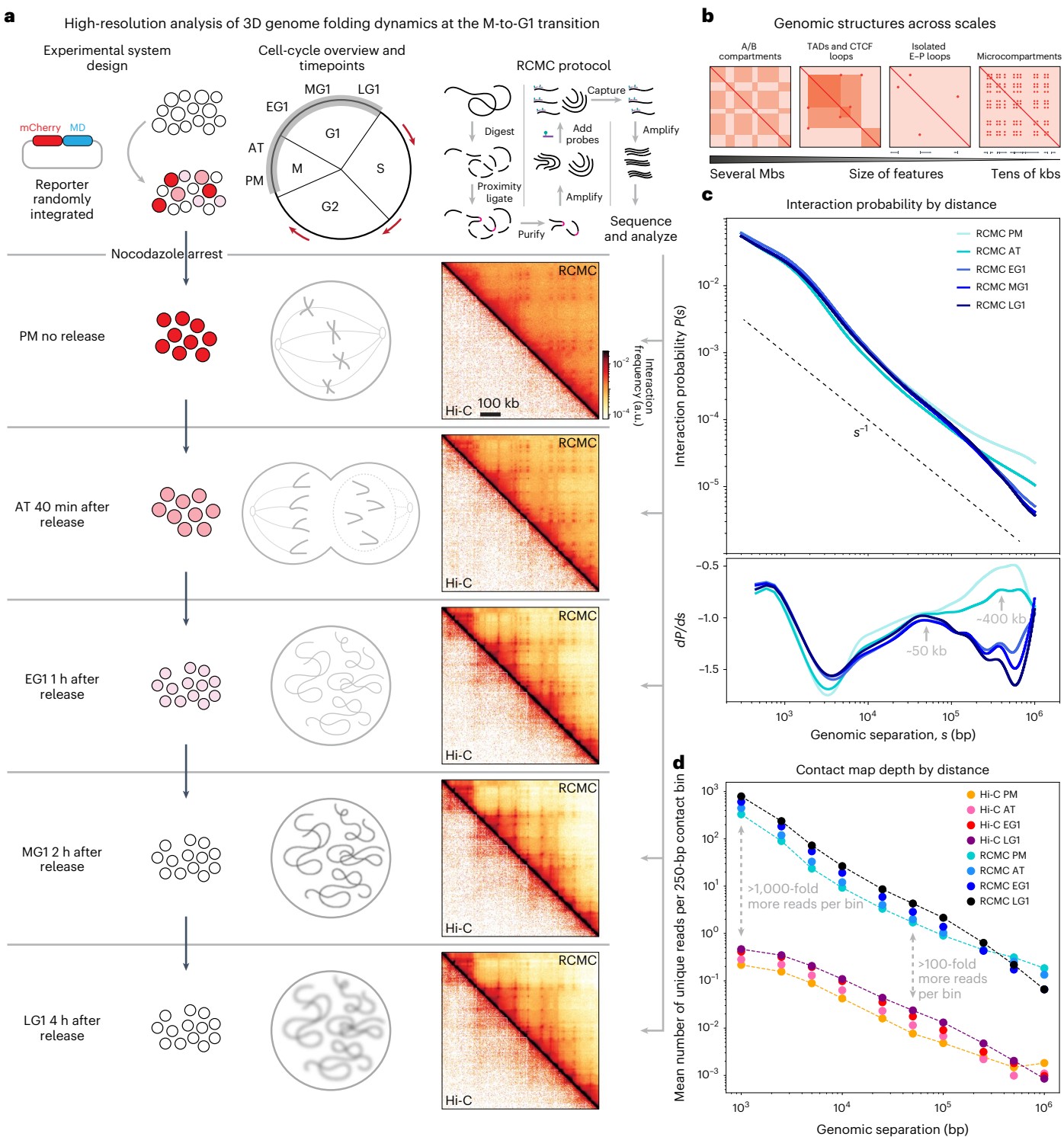

**Fig. 1 | RCMC deeply resolves 3D genomic architecture at the M-to-G1 transition. a**, Overview of the experimental system. As previously described[18], G1E-ER4 cells with an mCherry-tagged mitotic domain reporter are PM-arrested using nocodazole and flow-sorted after release to capture highly pure cell populations across five M-to-G1 time points: PM (no release), AT (25 min after release), EG1 (1 h), MG1 (2 h) and LG1 (4 h) (Extended Data Fig. 1a). The RCMC protocol[28] is applied to each of these cell populations; briefly, chromatin is chemically fixed, digested with micrococcal nuclease (MNase) and biotin-labeled before proximity ligation joins spatially proximal fragments. After enrichment for ligated interactions, fragments are library-prepped, amplified and region-captured to create an RCMC library that is sequenced, mapped and normalized to create contact matrices. **b**, Schematic representation of how A/B compartments, TADs, CTCF loops, E–P loops and microcompartments appear in contact maps across scales. **c**, Interaction probability curves comparing the interaction frequency at different genomic separations ($s$) for the five RCMC datasets. The first derivative of these $P(s)$ curves is shown at the bottom. **d**, The 3C data density in captured regions for RCMC versus Hi-C data from Zhang et al.[18]. Averaged counts for the number of unique reads across five captured regions are plotted for increasing interaction distances for all datasets at a 250-bp bin size.

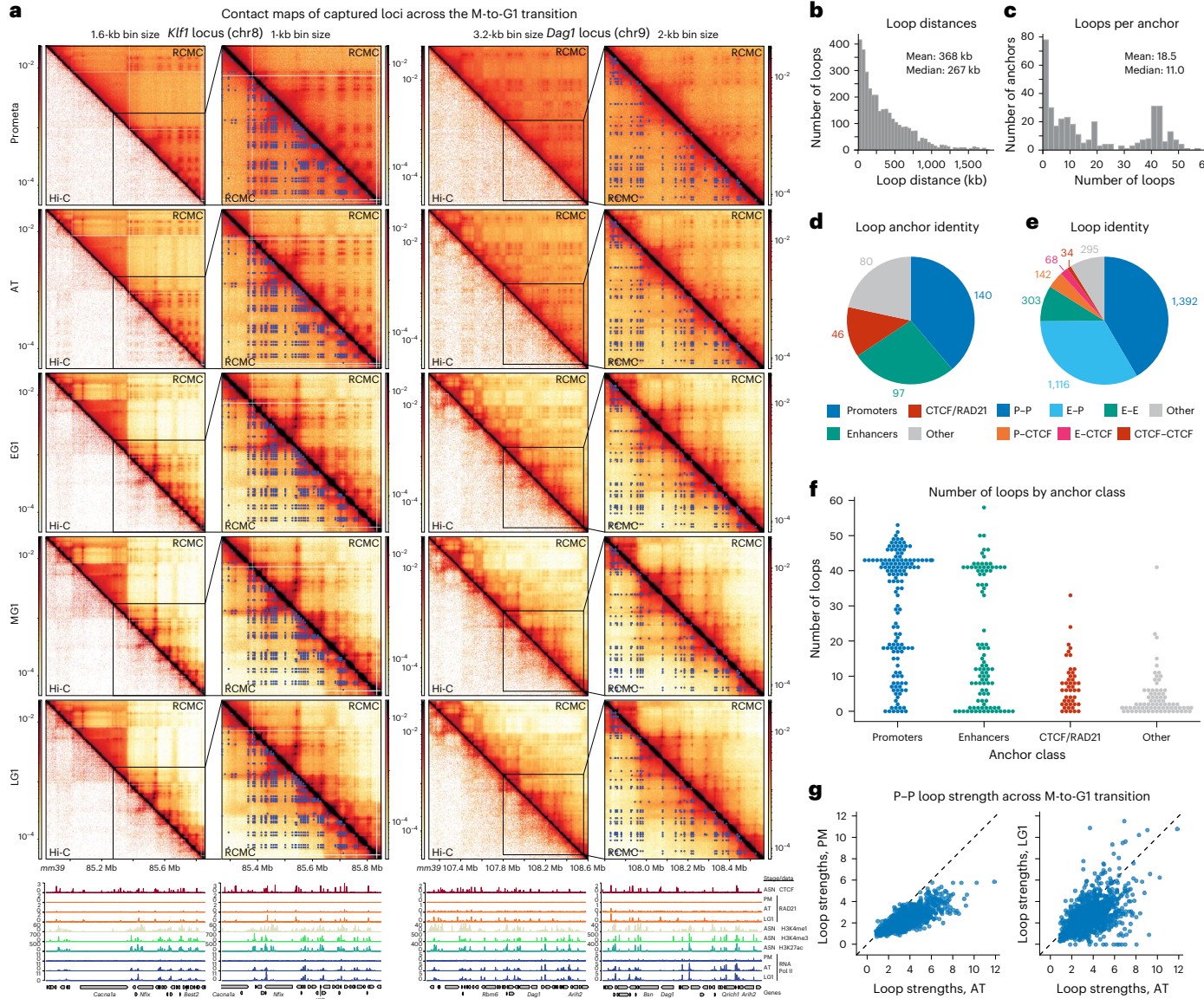

**Fig. 2 | RCMC finely resolves dynamically changing focal looping interactions.**
**a**, Contact map visualization of RCMC data at the *Klf1* (bin size: 1.6 kb, left; 1 kb, zoomed-in view) and *Dag1* (bin size: 3.2 kb, left; 2 kb, zoomed-in view) loci across the M-to-G1 transition, with Hi-C data[18] (left) and the superset of loops (right) shown below the diagonal. Genomic annotations and ChIP data (stage-specific and asynchronous) are shown at the bottom. **b,c**, Histograms of loop interaction distances (**b**) and the number of interactions formed by each annotated anchor (**c**). **d**, Pie chart of annotated loop anchors by their genomic identity, determined by chromatin features within 1 kb of anchor sites. Promoters were identified as annotated TSSs ± 2 kb, enhancers as nonpromoter regions with overlapping H3K4me1 and H3K27ac ChIP-seq peaks and CTCF/RAD21 as nonpromoter and nonenhancer sites with overlapping CTCF and RAD21 ChIP-seq peaks. Anchors with multiple overlapping genomic features were hierarchically classified into

a single classification, with promoters taking precedence, then enhancers and finally CTCF/RAD21. Anchors designated as 'other' do not overlap promoters, enhancers or CTCF/RAD21. **e**, Pie chart of annotated loops by the genomic identity assigned in **d**, with P designating promoters, E designating enhancers and CTCF designating CTCF/RAD21. **f**, Swarm plot of the number of interactions formed by each annotated anchor, separated by the genomic categories shown in **d**. **g**, Plots of individual P–P interaction strengths in the PM (left) and LG1 (right) conditions, plotted against the strengths in the AT condition (*x* axes). Strengths are calculated as the integrated observed loop signal divided by the expected background signal from local *P*(*s*) curves ('observed over expected'). 'Exclusive' P–P loops are shown; 186 P–P loops that overlap with CTCF at one or both anchors are removed and all subsequent loop pileups and quantifications of strength by loop identity similarly omit loops meeting both CRE and CTCF/RAD21 loop types.

across the M-to-G1 transition. As above (Fig. 2a), the nested CRE loops that comprise microcompartments peak in strength in AT, in part because of loss of background interaction from PM to AT, before gradually weakening during G1 (Fig. 3a, i–iv). Loop pileups (Fig. 3b and Supplementary Fig. 3b) and strength quantifications (Fig. 3c and Supplementary Fig. 3a) for each functional categorization revealed that CRE loops weaken relative to their background after peaking in AT, whereas CTCF-anchored loops are relatively weak in PM but monotonically strengthen to be stronger than CRE loops by G1. In G1, CTCF loops are ~3-fold stronger than CRE loops (Fig. 3c), which is similar

to interphase mouse embryonic stem cells[39]. Moreover, pileups of RCMC-detected loops in Hi-C data[26] recapitulate the transient spike in CRE loop strengths in AT and the gradual increase in CTCF-anchored loop strengths into G1, providing orthogonal validation of these trends (Supplementary Fig. 3c).

In our analyses thus far, we quantified loop strength as signal divided by background, as is standard in the field (observed/expected; Fig. 3b,c). However, as the expected strength, *P*(*s*), decreases with separation, this involves dividing by very small numbers for very distal loops and the *P*(*s*) itself strongly changes during the M-to-G1 transition

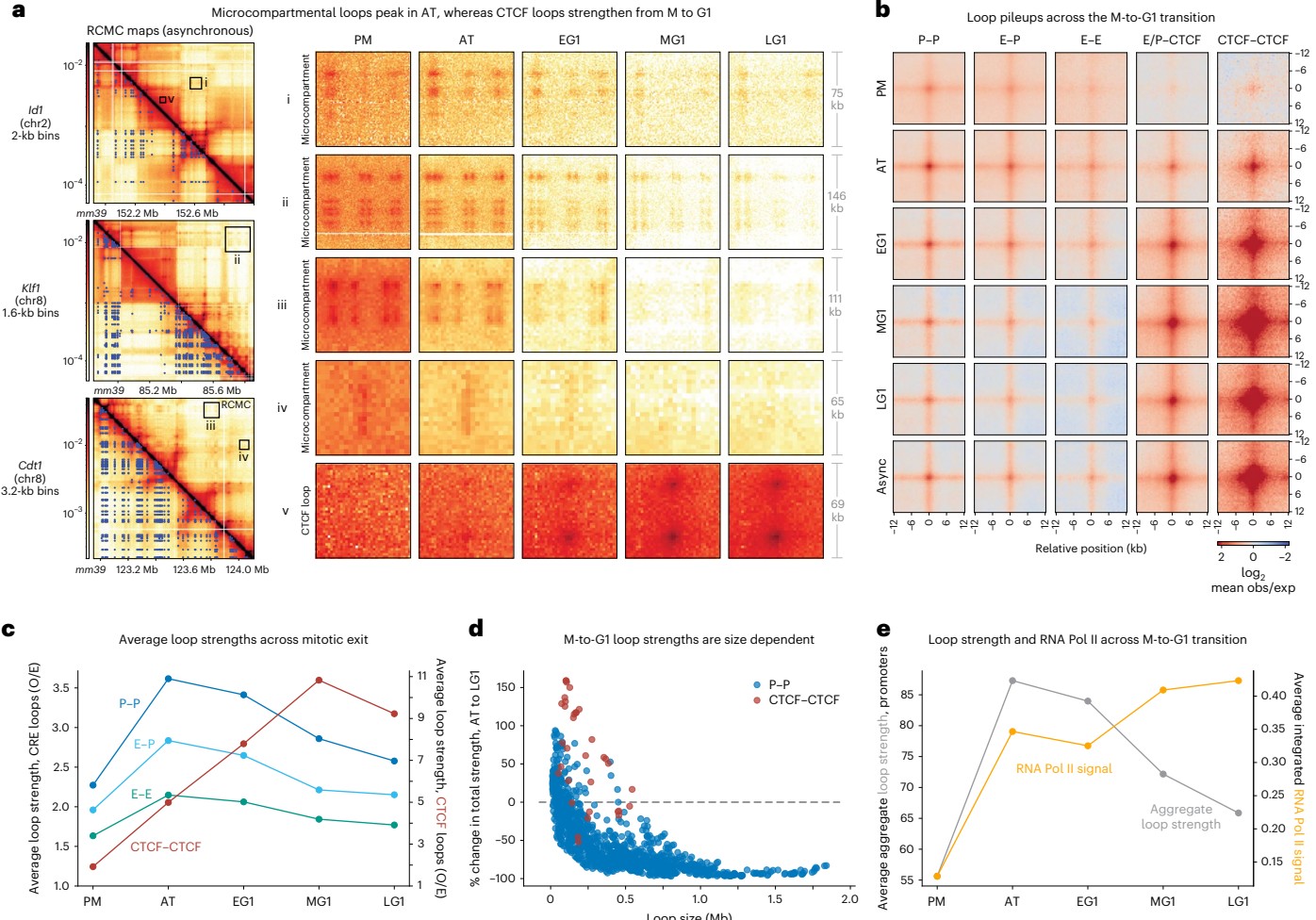

**Fig. 3 | The strength of CRE loops and microcompartments peaks in AT and weakens as cells enter G1 phase. a**, Asynchronous RCMC contact maps (left) at the *Id1*, *Klf1* and *Cdt1* regions with manually annotated interactions shown below the diagonal. Right, zoomed-in boxes shown in greater detail across the M-to-G1 transition using consistent color map scaling. Zoomed-in boxes show examples of microcompartmental CRE loops in i–iv and CTCF/RAD21 loops in v. **b**, APA plots of loops, separated to show P–P, E–P, E–E, E/P–CTCF and CTCF–CTCF loops across the M-to-G1 transition and for the asynchronous condition. Plots show a 24-kb window centered on the loop at 500-bp resolution and the loops plotted here and in all subsequent panels follow the 'exclusive' definition of loop identity the same as in Fig. 2g (CRE sites do not overlap with CTCF). **c**, Average loop strengths across mitotic exit, with CRE loop strengths on the left axis and CTCF/RAD21-anchored loop strengths on the right axis. Loop strengths were

calculated as 'observed over expected' signal, the same as in Fig. 2g. **d**, Change in loop strength across mitotic exit as a function of loop size. The percentage change in total loop strength for each P–P (blue) and CTCF–CTCF (red) loop from AT to LG1 is plotted on the *y* axis, while the loop size (or interaction distance) is plotted on the *x* axis. Total loop strengths are calculated as the observed signal without normalization for the expected signal. **e**, Promoter loop strengths (gray) and RNA Pol II signal (yellow) across mitotic exit, with loop strengths on the left axis and Pol II signal on the right. Loop strengths were calculated as the sum of all observed over expected loop strengths at each promoter and averaged across all promoters. RNA Pol II signal was calculated as the aggregate signal within 1 kb of each promoter-classified loop anchor, averaged across all promoters for each M-to-G1 stage.

(Fig. 1c). Moreover, for CRE loops, it is likely that it is the total loop strength—rather than the background-normalized strength—that is functionally important[39]. Therefore, we also quantified total loop strength without background normalization. Changes in total loop strengths across the M-to-G1 transition confirm our observations that most CRE loops peak in AT, whereas CTCF loops strengthen through G1. Notably, we now clearly observe that these dynamics are highly distance dependent, as the absolute strengths of all distal loops drops from AT to LG1 because of a systematic weakening of interactions far from the diagonal (Fig. 3d; compare to observed/expected strength quantifications in Supplementary Fig. 4); by contrast, some short-range CRE loops strengthen from AT to LG1. Thus, CRE looping dynamics show a clear distance dependence.

Lastly, we analyzed the relationship between microcompartments and transcription. Notably, we observed that the transient spike in

microcompartment strength coincides with a spike in RNA polymerase (Pol) II (Fig. 3e); this suggests a structure–function relationship, which we subsequently investigated in more detail.

**Transcriptionally spiking genes are associated with strongly peaking microcompartments at the M-to-G1 transition**
Our observation of the correlation between microcompartments and transcription during mitosis prompted a closer analysis of the relationship between the two and whether it may be responsible for a previously reported mitotic transcriptional spiking phenomenon[1,42–44] (Fig. 4a). Specifically, prior studies overturned the notion that transcription is entirely shut off in mitosis by demonstrating that many genes exhibit a transient hyperactive transcriptional state ('spike') near AT and EG1 (refs. 1,42–44) (Fig. 4a). However, a mechanistic explanation for this transcriptional spiking has been lacking. On the basis of the observed

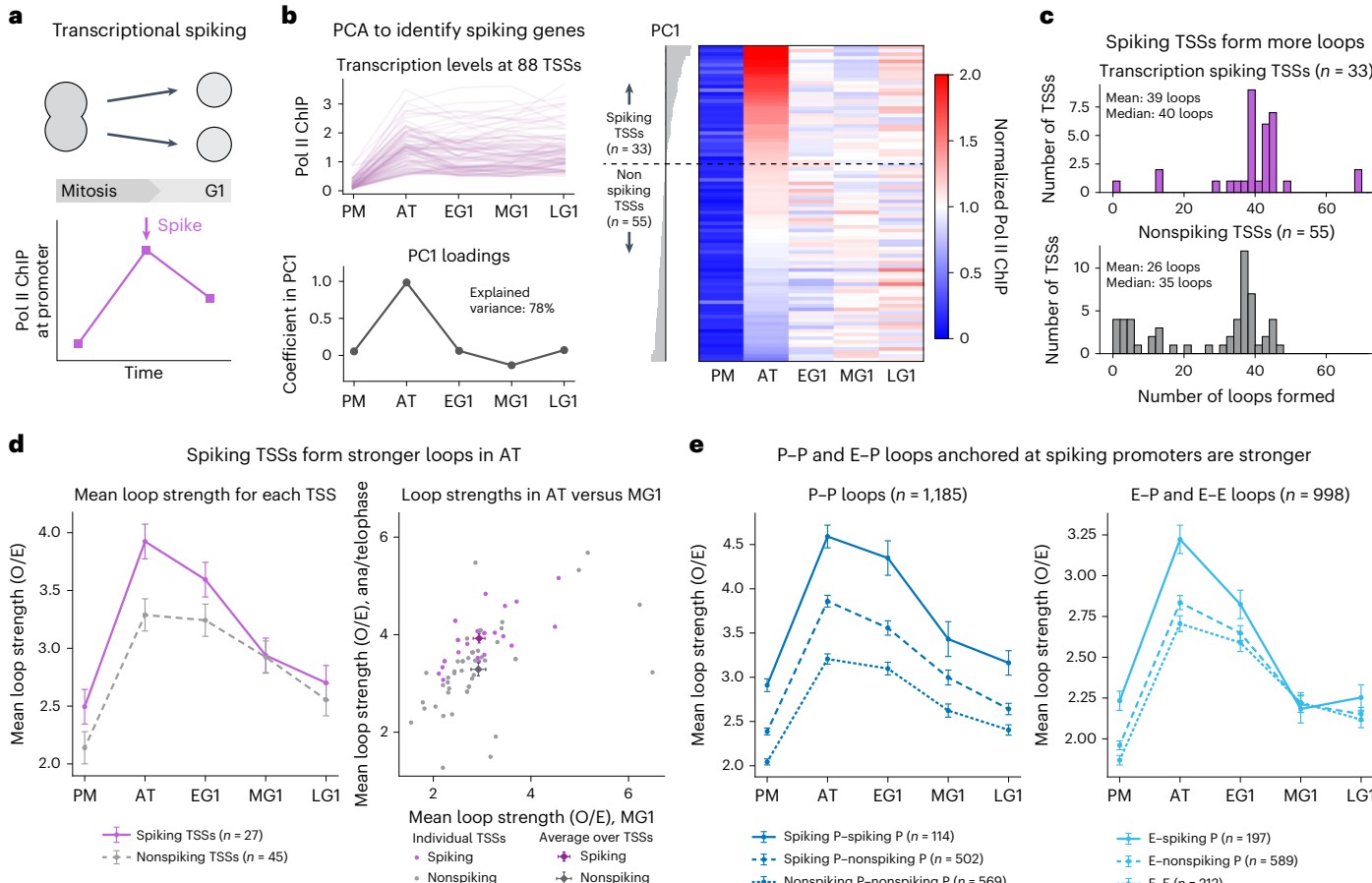

**Fig. 4 | Transcriptionally spiking genes are associated with strongly peaking microcompartments at the M-to-G1 transition. a**, Qualitatively, transcriptional spiking is characterized by activity during the M-to-G1 transition that exceeds the steady-state level observed later in G1. **b**, Top left, among the 88 active genes analyzed, Pol II ChIP levels at the TSSs exhibited varying behaviors over time. Bottom left, after normalizing each time series by its mean value in G1 and performing PCA, PC1 represented transcriptional spiking. Right, a heat map of the normalized Pol II ChIP values sorted by PC1 visually confirms PC1 as a quantifier of transcriptional spiking; TSSs with PC1 > 0 were identified as spiking. **c**, Histograms of the number of loops formed by transcriptionally spiking promoter anchors (top) and by nonspiking promoter anchors (bottom). **d**, Left, mean strength of loops anchored at spiking TSSs (purple solid line) and

nonspiking TSSs (gray dashed line) across time. Error bars indicate the s.e.m. Right, scatter plot of mean loop strength in AT versus mean loop strength in MG1 for all TSSs. Averages over all TSSs are shown, with error bars indicating the s.e.m. **e**, Left, average P–P loop strengths across mitotic exit, separated to show loops formed by two spiking promoters (solid line), two nonspiking promoters (lower dotted line) and one of each (middle dashed line). Loops shorter than 10 kb were excluded from strength quantification to allow adequate buffer for correction by local background calculation. Error bars indicate the s.e.m. Right, Average E–P and E–E loop strengths across mitotic exit, separated to show enhancer loops formed with one spiking promoter (solid line), one nonspiking promoter (middle dashed line) and another enhancer (lower dotted line). Error bars indicate the s.e.m.

aggregate relationship between microcompartmentalization and transcription (Fig. 3e), we hypothesized that microcompartmentalization of CREs in mitosis could be the mechanism driving transiently spiking transcription in mitosis. To test this, we analyzed transcriptional spiking together with microcompartment formation dynamics at individual genes.

We first identified spiking genes on the basis of the temporal behavior of their RNA Pol II chromatin immunoprecipitation (ChIP) levels during the M-to-G1 transition[42]. For each active gene located in the RCMC capture regions, we calculated the normalized Pol II ChIP signal within a 2.5-kb window centered on the TSS at all five time points and performed principal component analysis (PCA) (Fig. 4b). The first principal component (PC1) represents a peak in transcription levels during AT and explains 78% of the variance between the genes (Fig. 4b). We subsequently labeled genes with positive PC1 values as 'spiking' and genes with negative PC1 values as 'nonspiking' (Fig. 4b), resulting in 38% of active genes being identified as spiking (Supplementary Fig. 5a).

To assess their relationships to microcompartments, we intersected these two classes of genes with microcompartment anchors

(Supplementary Fig. 5b–e). Most genes captured in ROIs (64%) overlap microcompartment anchors (Supplementary Fig. 5b), with spiking promoters forming a sizeable fraction of microcompartment promoter anchors (Supplementary Fig. 5c) and a large fraction of promoter-anchored microcompartment loops formed by spiking promoters (Supplementary Fig. 5d). Furthermore, transcriptionally spiking TSSs form more loops (Fig. 4c), longer loops (Supplementary Fig. 5f) and stronger loops (Fig. 4d) compared to nonspiking TSSs. The difference between the average loop strengths formed by spiking versus nonspiking TSSs is most pronounced during AT and vanishes by MG1 (Fig. 4d). This relationship also holds for individual TSSs, as the microcompartment strengths of spiking genes peak more strongly during the AT, despite wide variability in loop strength (Fig. 4d). We similarly found that transcriptional spiking at microcompartment anchors is correlated with stronger loops for both P–P and E–P loops (Fig. 4e). Collectively, these results are consistent with a dynamic structure–function relationship in which microcompartment formation and establishment is upstream of transcriptional activity during the M-to-G1 transition.

## Condensin depletion sharpens A/B compartments but not microcompartments

Previous work has shown that A/B compartments, formed by large continuous blocks of epigenetically distinct chromatin (hundreds to thousands of kilobases), strengthen after loss of cohesin-mediated loop extrusion in interphase[45,46]. Recently, to generate a loop-extrusion-free chromatin environment while minimizing the confounding effects of transcription and most transcription factors, we depleted SMC2, a common subunit of condensins I and II, in PM[17]. Condensin depletion led to strong gains in A/B compartmentalization and in low-connectivity CRE loops in mitosis[17], which now prompted us to explore how microcompartments self-organize without condensins.

We applied RCMC to the same experimental system[17] (Fig. 5a) to generate RCMC data in PM mitotic chromosomes across five SMC2 depletion time points, with deeply resolved contact maps for the 0-h, 1-h and 4-h depletion time points (Fig. 5b) and sparser datasets for the 0.5-h and 8-h time points (Supplementary Figs. 6–10). Following near-complete condensin depletion (Supplementary Fig. 9a,b), we observe visually striking strengthening of contrast in the checkerboard pattern characteristic of A/B compartmentalization (Fig. 5b). The strengthening of large-scale A/B compartments matches what we previously observed[17], thus validating our RCMC maps.

Next, we explored the effects of condensin depletion on microcompartments by quantifying individual loop strengths (Fig. 5c and Supplementary Fig. 9e) and generating pileups (Fig. 5d and Supplementary Fig. 9f) averaged across all loops. While we did observe strengthening of several CRE loops after condensin depletion (Fig. 5c), the changes to microcompartment loops upon condensin depletion were minor on average (Fig. 5d). In contrast, analysis of large A/B compartments further confirmed that they sharply increase in strength over time[17] (Fig. 5e,f) without strongly affecting microcompartments. The divergence in the effects of condensin depletion on large A/B compartments versus microcompartments may be partially explained by microcompartments rarely crossing compartmental boundaries. In concordance with their formation at CREs, microcompartment anchors primarily fall within A compartments (61%), with 58% of microcompartment loops being intra-A, 24% being intra-B and 18% being inter-A/B compartment interactions. Collectively, the condensin depletion RCMC data point toward mitotic loop extrusion acting more antagonistically toward A/B compartments formed by larger (hundreds to thousands of kilobases) blocks than toward microcompartments formed by smaller blocks (~1–10-kb loop anchors), suggesting that the relative sensitivity of compartments to loop extrusion may be size dependent.

In summary, we find that large A/B compartments and microcompartments appear to be at least partially mechanistically separable, as they exhibit temporally distinct formation dynamics upon mitotic exit and distinct sensitivities to loss of condensin in mitosis (Fig. 5g). To further explore their mechanistic basis, we turned to experimentally constrained 3D polymer simulations.

## Loop extrusion activity, chromatin affinity and compaction regulate microcompartments

To investigate the biophysical factors underlying formation, maintenance and dynamics of microcompartments, we developed a polymer model incorporating major mechanisms of chromatin organization[47]. We modeled loop extrusion by dynamically exchanging SMC complexes (condensin and cohesin) that bind to the chromatin fiber and perform two-sided extrusion before unbinding (Fig. 6a, top)[37,48–50]. We also modeled affinity-based homotypic interactions for three types of chromatin (A, B and C) to capture the formation of large A and B compartments and small microcompartments (denoted as type C, for CRE anchors; Fig. 6a, bottom left). We specifically modeled the *Dag1* locus, which we embedded in a larger polymer chromosome confined to a sphere at a chosen volume density (Fig. 6a, bottom right).

To understand how affinity-based interactions and loop extrusion influence microcompartmentalization, we performed parameter sweeps without extrusion-stalling CTCF sites and computed steady-state contact maps (Supplementary Table 2). As microcompartment affinity, $\epsilon_C$, was increased, microcompartments became more visible and more sharply defined (up to ~8-fold difference from bottom to top row in Fig. 6b) because stronger affinity promotes longer-lived interactions (Fig. 6b, Supplementary Fig. 11 and Extended Data Fig. 9a). Furthermore, distinct microcompartments formed in the presence or absence of a weaker background of larger-scale A/B compartments and microcompartments were insensitive to their genomic positioning relative to A/B compartment segments (Extended Data Fig. 9b,c). The simulations indicate that microcompartments can be formed by sufficiently strong affinities between small chromatin segments and their prominence in contact maps may be tuned largely independently of larger A/B compartments.

Intriguingly, the appearance of microcompartments in the model was also influenced by loop extrusion dynamics. As cohesin and condensin have different residence times, we explored this effect by modulating the extruder residence time, $\tau_{res}$, at a fixed linear density of extruders. We found that faster turnover (that is, shorter residence time) partially or fully suppressed microcompartmentalization, even for the strongest microcompartment affinities (Fig. 6b). Thus, a longer residence time of loop extruders, such as condensin II, results in stronger microcompartments (Fig. 6b). This contrasts with previous experimental and computational findings for larger

---

**Fig. 5 | Condensin depletion sharpens A/B compartmentalization while preserving microcompartments. a**, Overview of the experimental system. As previously described[17], G1E-ER4 cells with mCherry-tagged SMC2–mAID are PM-arrested using nocodazole and treated with auxin to induce rapid depletion of SMC2 for 0, 0.5, 1, 4 and 8 h at the end of an 8-h (all but the 8-h depletion) or 15-h (the 8-h depletion) nocodazole arrest. SMC2 degradation eliminates both condensins I and II. Cells are then RCMC-crosslinked, sorted for M-phase purity and processed into sequencing libraries using the RCMC protocol. **b**, RCMC contact maps comparing the *Klf1* (plus zoomed-in boxes), *Dag1*, *Id1* and *Cdt1* loci following 0, 1 and 4 h of SMC2 degradation. Interaction annotations generated from the M-to-G1 RCMC data are overlaid below the diagonal. Right, contact intensity scaling. **c**, Plots of individual loop strengths in the 1-h (top) and 4-h (bottom) depletion conditions, plotted against the strengths in the control 0-h depletion condition (*x* axes), for P–P loops (left) and CTCF/RAD21–CTCF/RAD21 loops (right). Loops were defined by their exclusive identities (no CRE and CTCF overlap) and strengths were calculated as the observed over expected signal. **d**, APA plots of called interactions, separated to show exclusively defined P–P, E–P, E–E, E/P–CTCF and CTCF–CTCF loops across SMC2 depletion. Plots show a 20-kb window centered on the loop at 500-bp resolution. **e**, Compartmentalization signature for the 4-h SMC2 depletion condition at the

*Dag1* locus. Eigenvector decomposition of interaction frequencies is shown above the contact map, with transition states between positive and negative values noted as black lines overlaid atop the RCMC map. The 4-h depletion condition is shown above the diagonal while the control 0-h depletion is shown below the diagonal. Of the 363 annotated microcompartment anchors across all target loci, 220 (61%) lie within A compartments and 39% lie within B compartments. Of the 3,350 annotated microcompartment loops, 1,941 (58%) are intra-A interactions, 803 (24%) are intra-B interactions and 606 (18%) are inter-A/B interactions. **f**, Saddle plots of progressive compartmentalization across the 0-h, 1-h and 4-h depletion conditions at the *Klf1*, *Dag1*, *Id1* and *Cdt1* loci. Track showing the strengths of the two compartments and their transition point, in which B-compartmental regions (for example, low eigenvector values) are shown toward the bottom and left of the track while A-compartmental regions (for example, high eigenvector values) are shown on the top and right. Each track is ordered by the eigenvector component values for the specified locus in the indicated treatment condition. **g**, Interaction probability curves comparing the interaction frequency at different genomic separations (*s*) for the five condensin depletion datasets. The first derivative of these *P*(*s*) curves is shown at the bottom.

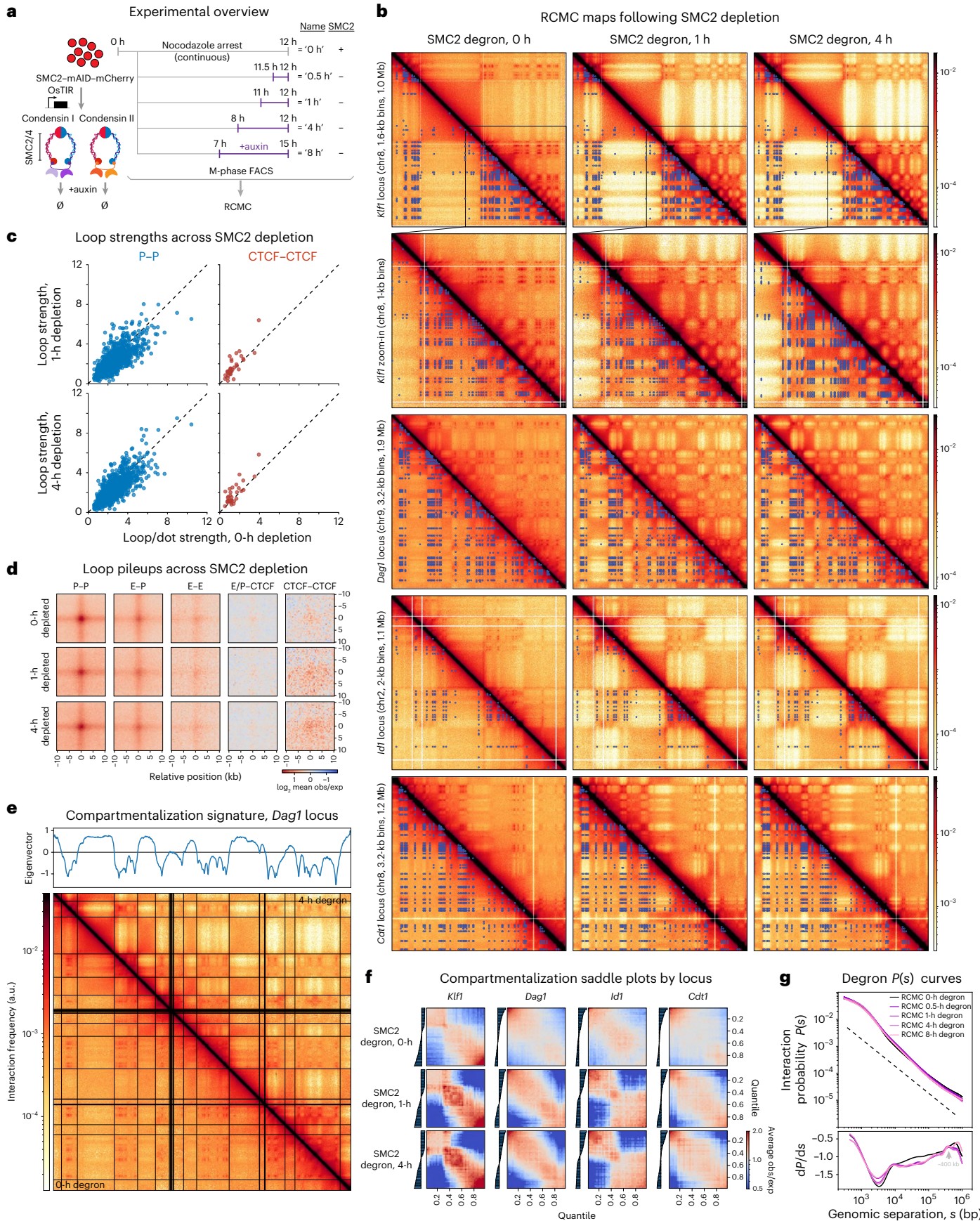

**a** Experimental overview

**b** RCMC maps following SMC2 depletion

**c** Loop strengths across SMC2 depletion

**d** Loop pileups across SMC2 depletion

**e** Compartmentalization signature, *Dag1* locus

**f** Compartmentalization saddle plots by locus

**g** Degron *P*(s) curves

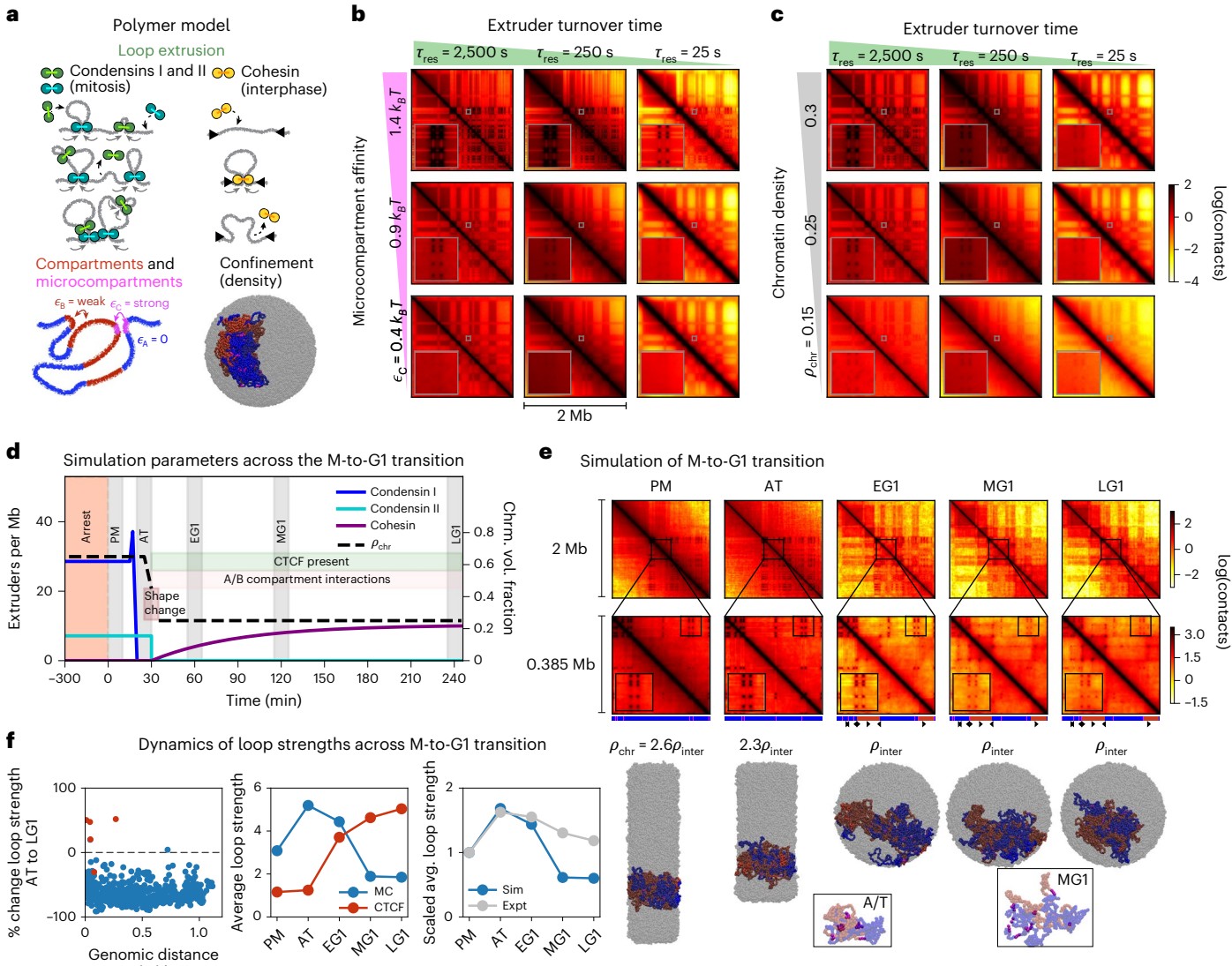

**Fig. 6 | Polymer simulations of chromosomes demonstrate how loop extrusion, interaction energy and polymer properties may govern microcompartmentalization throughout the M-to-G1 transition.**
**a**, Illustrations of key components of the simulation model. Top left, condensins I and II (green and turquoise, respectively) dynamically bind and unbind to the chromatin fiber (gray) and extrude chromatin polymer loops. Condensin I has a relatively short residence time, $\tau_{res}$, which results in the formation of small loops nested within large loops formed by condensin II. Top right, cohesin (yellow) extrudes loops and may stop when it encounters correctly oriented CTCF (black arrowheads). Bottom left, the chromatin fiber is a block copolymer with three types of blocks, which self-interact with affinities given by the interaction energies, $\epsilon_i$. Bottom right, the *Dag1* region (colored) is simulated as part of a larger polymer chromosome (gray), which is confined to a sphere. **b,c**, Contact maps from steady-state simulations of the *Dag1* region for different loop extruder residence times, $\tau_{res}$ (decreasing from left to right columns), and microcompartmentalization affinities, $\epsilon_C$ (decreasing from top to bottom rows; **b**), or different polymer volumetric densities, $\rho_{chr}$ (decreasing from top to bottom; **c**). Linear density of loop extruders, $1/d$, was fixed at one extruder per 100 kb in these simulations. Small gray boxes denote regions magnified in insets. **d**, Summary of simulation model of chromosome organization throughout the M-to-G1 transition. Lines show the linear densities of condensins I and II and

cohesin, as well as a 2.6-fold decrease in polymer density through the M-to-G1 transition. Gray regions indicate the time during which data were collected for annotated cell-cycle phases. The red region indicates initial equilibration of the simulation modeling PM arrest, as further described in the text. **e**, Bottom, contact maps from various times in the M-to-G1 transition simulations with corresponding simulation snapshots (bottom). The middle row displays zoomed-in views of the region indicated in the top row. Insets within this row show the 40-kb × 40-kb region of the contact map indicated by the small black box, showing the dynamics of microcompartmental contacts throughout the transition. Compartment structure and CTCFs are indicated for this region beneath the maps. Images show snapshots of polymer simulation with a single *Dag1* region colored. Boxed images at the bottom show snapshots of a 0.385-Mb segment of the *Dag1* region with A and B monomers (blue and red) made transparent to highlight microcompartments (magenta). **f**, Quantification of percentage change in loop strength of simulated microcompartments from AT to LG1 as a function of loop size (left) and average microcompartment and CTCF loop strengths (middle) throughout the M-to-G1 transition, the same as in Fig. 3c,d. Right, microcompartment loop strengths scaled by mean PM loop strength for simulations (blue) and corresponding P–P, E–P and E–E loops in the *Dag1* region in experiments (gray).

A/B compartments, which can be erased by extrusive cohesin with a long residence time (for example, because of WAPL loss)[45,51,52]. Instead, for microcompartments, the increase in total extrusion activity (that is, extrusion steps per unit time per megabase) induced by

faster extrusion turnover can erase or suppress microcompartmentalization (Extended Data Fig. 9f). The notion that extrusion activity is important in suppressing microcompartmentalization is further supported by simulations with different extruder linear densities and

velocities (Extended Data Fig. 9a,d–f). Furthermore, in simulations with quenched static loops instead of active extrusion, microcompartment strengths were much stronger and less sensitive to the extrusion parameters used to generate the loops (Extended Data Fig. 9g). Intuitively, because extruding through a microcompartmental interaction tends to disrupt it, increasing extrusion activity in the model generally weakens microcompartments. Nonetheless, disruption of microcompartments by extrusion in simulations contrasts with the minimal effect of the condensin degradation experiments (Fig. 5), suggesting that additional factors may maintain chromosome structure during the M-to-G1 transition (for example, Extended Data Fig. 9h,i).

Because chromatin density changes ~1.5–3-fold through the M-to-G1 transition[53,54], we simulated systems with different polymer densities (chromosome compaction). We observed that microcompartments were more prominent in systems at higher density (Fig. 6c). In denser systems, such as compacted mitotic chromosomes, the configurational entropy of the polymer is decreased because of the decrease in accessible volume. This reduces the entropic penalty of microcompartment formation, thus favoring the formation of microcompartments in more densely compacted chromosomes. Across all simulated densities, increased loop extrusion activity suppressed microcompartments, although the effects of extrusion were smaller at higher densities (Fig. 6c and Extended Data Fig. 9i). The effect of density on microcompartment strength is highly nonlinear; for a twofold increase in density, microcompartment strength increased by ~30% (Fig. 6b), and strength could be increased ~6-fold through another twofold density increase (Extended Data Fig. 9a). These simulations indicate that chromatin polymer density can act as a global physical regulator that influences microcompartment formation through both graded and sharp changes.

Together, our simulations uncovered that three factors influence the strength of microcompartments. While homotypic affinities between the anchors and higher chromosome density make them stronger, loop extrusion generally weakens microcompartments, with extruders that turn over faster affecting microcompartments to a greater degree.

### Chromatin density and loop extrusion govern microcompartments in simulations of the M-to-G1 transition

An interplay of affinity, extrusion and density affects microcompartment strength in a steady state; however, it was unclear how these factors collectively govern microcompartments in a time-varying context, such as the M-to-G1 transition. We implemented the polymer simulation components depicted in Fig. 6a, with time-varying, experimentally estimated extrusion and density parameters to model the progression from PM arrest to LG1 (Fig. 6d, Methods and Supplementary Table 3), while holding microcompartmental affinity constant. Timescales were calibrated similarly to simulations in Gabriele et al.[55], using live-cell locus tracking data to integrate polymer dynamics and loop extrusion and model the passage of time between cell-cycle phases.

The simulation proceeds with (Fig. 6d) (1) initialization and equilibration of the chromatin polymer within cylindrical confinement[16], with microcompartmental affinities and loop extrusion by condensins I and II during PM arrest; (2) PM; (3) condensin I increase after PM, before gradual removal[7,56,57]; (4) AT, during which the confining cylinder shortens and widens and polymer density decreases[53,54,58]; (5) condensin II removal[7,19,59,60], addition of CTCF and A/B compartment affinities[18], onset of a gradual crossover from cylindrical to spherical confinement at a lower polymer density[61] and onset of increasing cohesin[7,18]; (6) EG1; (7) MG1; and (8) LG1.

Contact maps for simulated chromosomes for each experimental RCMC time point in the M-to-G1 transition showed a complex and evolving architecture, as observed in the experiments (Fig. 6e). Focal enrichments indicating microcompartments were visible across all time points and they were particularly strong in PM, AT and EG1.

A/B compartments, TADs and CTCF–CTCF loops emerged in EG1 and strengthened through G1 as cohesin was loaded and the chromatin polymer reequilibrated. Notably, simulations revealed that microcompartments are often formed through multiway interactions, that is, focal enrichments typically resulted from simultaneous microphase separation of ~5 microcompartmental (C-type) anchors (Supplementary Fig. 12).

With the chosen temporal evolution of density and extrusion dynamics, microcompartmental loops peaked in strength in AT, whereas CTCF loops uniformly increased (Fig. 6f), as in the RCMC experiments (Fig. 3c,d). Scaled loop strengths in simulations quantitatively matched experimental measurements for PM, AT and EG1 (Fig. 6f), after which the assumption of constant interaction affinity likely fails as more and more transcription factors and other DNA-binding proteins rebind to CREs[62]. Our simulations indicated that microcompartments can be generated by block copolymer microphase separation, a polymer-based mechanism distinct from protein-based liquid–liquid phase separation[28,33,34,45,63,64]. In addition, we found that microcompartments can dynamically change through biophysical mechanisms that act and change during the M-to-G1 transition, even if CRE affinities remain constant.

Simulations suggested that trends in the observed strengths of microcompartments largely, but not exclusively, emerged because of the difference in chromatin densities between mitosis and G1. In simulations in which chromosome density was held constant, microcompartments were stronger in G1 (Supplementary Fig. 13). As observed in experiments (Fig. 5), loop extrusion is not necessary to form microcompartments (Extended Data Fig. 10a–c). However, extruders can diminish microcompartments in simulations, as observed with shorter residence times (faster turnover) or more loop extruders (Extended Data Fig. 10d). Furthermore, the timing of condensin I removal is responsible for the strengthening of microcompartments in AT relative to PM. In simulations in which condensin I was removed during AT, microcompartments instead peaked in strength during EG1 (Extended Data Fig. 10f); this can be remedied by reducing condensin I turnover in AT (Supplementary Fig. 14a). Likewise, condensin II removal facilitates EG1 microcompartmentalization (Supplementary Fig. 14b). The strength of extrusion's effects on microcompartments depends on the speed of polymer dynamics relative to loop extrusion; faster polymer dynamics leads to a more muted impact of extrusion (Extended Data Fig. 10e). Otherwise, there is little or no dependence on other model assumptions, including changes in the shape of the confinement and A/B compartment interactions (Supplementary Fig. 15). Overall, using parameters mostly determined from experiments with minimal adjustment, the model generally reproduces experimental contact maps and loop strengths from mitosis to G1.

In summary, our simulation results showed that microcompartments are regulated by at least three distinct biophysical factors: homotypic affinity, chromatin density and loop extrusion activity. Each of these factors, in turn, can be regulated by distinct mechanisms and pathways.

## Discussion

Chromosomes are dramatically reorganized across the M-to-G1 transition. Prior work using Hi-C showed that all interphase 3D genome structural features, including A/B compartments, TADs and loops, are lost in mitosis and gradually reformed during G1 (refs. 1,16,18–22,24,25). Here, we applied RCMC[28] to the M-to-G1 transition[18] and achieved ~100–1,000-fold higher depth (Fig. 1d) than achieved by Hi-C[18]. Our RCMC maps are consistent with Hi-C at coarse resolution, but unexpectedly reveal a previously unobservable layer of 3D genome structure at fine resolution, most notably microcompartments that are present in mitosis (Fig. 2a). We observed that not only do many CREs come together to form microcompartments in both PM and AT but also most CRE interactions peak in strength in AT before weakening upon G1 entry

(Fig. 3). Thus, we showed that microcompartments are unique amongst 3D genome features as being retained during mitosis.

Nevertheless, we do note experimental limitations. Like all genomics assays, RCMC provides time-averaged and ensemble-averaged snapshots of pairwise interactions in cell populations. Thus, RCMC cannot resolve multiway structures in single cells, quantify absolute looping probabilities[39] or measure loop lifetimes[55]. We estimate the cell purification method to result in ~98% pure PM and AT populations (Extended Data Fig. 1), which includes ~2% contamination. However, because microcompartments are stronger in mitosis than in G1, we can exclude the possibility of mitotic microcompartments being because of interphase contamination because interphase contamination would only weaken microcompartment strength (Fig. 3d).

The presence of microcompartments in mitotic chromosomes provides insight into the mechanism of microcompartment formation because the formation mechanism must be compatible with the state of the genome in mitosis. As transcription is largely shut off in PM and RNA Pol II is largely absent, their presence in PM confirms that microcompartments do not require transcription to form. This is consistent with prior work that found only modest quantitative changes to CRE loops upon transcription and RNA Pol II perturbations[1,22,26,28,30,65]. Other candidate mediators of microcompartment formation include chromatin state and histone modifications, as well as chromatin and transcriptional regulators. For instance, promoters and, to a lesser extent, enhancers retain some chromatin accessibility during mitosis[41]. Furthermore, CREs retain H3K4me1/3 in PM and H3K27ac to some extent[20,66,67] and H3K27ac likely has a mitotic bookmarking role[21,66]. Thus, it appears that microcompartments reflect the epigenetic state of mitotic chromosomes, although more work is required to understand whether the relationship is correlative or causal. Moreover, while transcription factors were historically thought to be absent from mitotic chromosomes[12,13], recent work found that some factors remain bound to mitotic chromosomes and, thus, may also serve a mitotic bookmarking function. These include SOX2 (refs. 68,69), TBP (ref. 70), BRD4 (ref. 66), ESRRB (ref. 71), NR5A2 (ref. 71), GATA1 (ref. 72) and many others[12,13,73]. Thus, putative mitotic bookmarking proteins are also candidate mediators of microcompartment formation. Lastly, we speculate that, rather than being fully mediated by a single factor, microcompartments most likely form through a 'strength in numbers' mechanism involving the combined affinity-mediated interactions of many factors.

Polymer modeling provided further mechanistic insight and showed that microcompartmentalization is largely controlled by three characteristics: homotypic affinity of microcompartment anchors (such as CREs), dynamics of loop extrusion and chromatin density. While it is unsurprising that stronger affinity leads to stronger microcompartments, our simulations revealed unexpected effects of loop extrusion and density. For loop extrusion, 'extrusion activity' $\left(\frac{\text{Extrusion steps}}{\text{Time} \cdot \text{Mb}}\right)$ appears to be the key parameter for short-distance interactions between microcompartments. Each time an extruder, such as condensin or cohesin, extrudes through a microcompartment anchor, the extruder can disrupt microcompartmental interactions by bringing other chromatin segments into contact with the microcompartment anchor, regardless of the segments' affinities for each other. Thus, the collective effect of the number of extruders, their residence time and processivity governs the stability of microcompartments (Fig. 6b,c and Extended Data Fig. 9a,d–f). This observation partially contrasts with previous findings for large A/B compartments, which weaken when increasing extruder residence time through WAPL depletion[45,51,52]. This contrast may be because of differences in the polymer dynamics of the shorter chromatin segments between microcompartments and the longer segments spanning A/B compartments[74]. Altogether, our observations strengthen the notion that microcompartments and larger A/B compartments may be differentially modulated, even though the underlying biophysical mechanisms (affinity with modulation by loop extrusion) are similar.

Our simulations also revealed that chromosome density and compaction have an unexpectedly large role: a twofold change in density, which approximately matches the difference between mitotic and interphase chromosomes[53,54], is sufficient to go from nearly absent to very strongly visible microcompartments (for example, Fig. 6c, bottom versus top rows). Physically, microcompartment formation is favored by enthalpy but disfavored by entropy. Microcompartment formation reduces the configurational entropy of the polymer but the spatial constraints introduced through compaction reduce this entropic cost, thereby promoting microcompartment formation. Thus, while the presence of microcompartments in mitotic chromosomes was unexpected, their presence is consistent with polymer modeling; microcompartment formation in PM is facilitated by high compaction (Fig. 6c) and telophase likely provides a uniquely favorable environment because of the combination of very low extrusion activity[7,19,56] and high compaction[53,54] (Fig. 6d), thus explaining why microcompartments peak in strength in AT (Figs. 3 and 6e,f). This model also predicts that perturbations that affect density (for example, osmotic shock) may affect compartments, and that cell types with higher chromatin density (for example, smaller nuclei) may form stronger compartments. This may also help explain the modest effects of interphase cohesin depletion on E–P interactions[28,75]; because cohesin depletion simultaneously decreases chromosome compaction and density[45,52,55,76] and decreases extrusion activity, these negative and positive effects on microcompartments may roughly cancel out, thus explaining a relatively modest effect overall[28,75]. Indeed, two preprints that came out after our preprint found that around half of CRE loops form without cohesin during the M-to-G1 transition, while around half are cohesin dependent[77,78].

Therefore, chromosome compaction and the lack of loop extrusion by cohesin emerge as leading factors for stronger microcompartmentalization in mitosis. Furthermore, the only consistent models that we found had slow extrusion dynamics after PM (Fig. 6d–f and Supplementary Fig. 14). This finding hints at the possibility that mitotic extrusion dynamics after PM may be rather subtle, as high extrusion activity of condensin I would weaken microcompartments (Extended Data Fig. 10d,f). Given the modest effect of condensin depletion on submegabase contacts[17] (Fig. 5g), low condensin activity could reflect a relatively low abundance of condensin I in mouse erythroid cells. Alternatively, the lack of changes in microcompartment strength upon condensin depletion could also suggest that extrusion activity by condensins is diminished during later stages of mitosis. Thus, the loops of mitotic chromosomes may be fully extruded by the end of PM with comparably less extrusion activity later.

Our polymer model (Fig. 6a,d) reproduces the key features of 3D genome folding during the M-to-G1 transition, including gradual formation of A/B compartments, TADs and CTCF loops, as well as microcompartments that peak in AT (Fig. 6e,f), but there are several limitations. These include uncertainty about how key parameters change from mitosis to G1, including microcompartment and A/B compartment affinities, extrusion parameters, overall solvent conditions and how condensin governs chromatin volume fraction. This type of uncertainty may partially explain the failure of the simulation model to recapitulate the condensin degron experiments. Furthermore, although polymer dynamics were calibrated from live-cell interphase chromatin dynamics[55], it is possible that polymer dynamics during mitosis are substantially different. In turn, loop extrusion by condensins could have a smaller than expected inhibitory effect on microcompartmentalization in cells (Extended Data Fig. 10e). Additionally, we did not explore the contributions of other mechanisms thought to be involved in A/B compartment formation, such as interactions with nuclear bodies (for example, the lamina, nucleoli and speckles)[79,80], chromatin–chromatin crosslinks (for example, HP1)[81,82] and active polymer dynamics[83–85], which might variably facilitate or hinder microcompartmentalization.

The same mechanism of block copolymer microphase separation[28,33,34,45,63,64,86,87] appears to explain compartmentalization across scales; large blocks result in A/B compartments[45,88,89], kilobase-sized blocks result in microcompartments[28,90,91] and introducing both results in coexisting A/B compartments and microcompartments (Fig. 6). This raises the question of whether microcompartments[28] and the active larger A compartment[89] are formed by the same molecular factors but at different scales. Several observations from our study suggest that they may be at least partially distinct. Firstly, microcompartments were strongly visible in mitotic chromosomes, whereas A/B compartments were absent (Figs. 2 and 3). Secondly, condensin depletion led to strong A/B compartmentalization in PM without strongly affecting microcompartments (Fig. 5). Thirdly, in simulations, we could largely tune A/B compartment strength and microcompartment strength independently, without them strongly affecting each other (Fig. 6, Extended Data Fig. 9a–c and Supplementary Fig. 15b–d). Lastly, increasing extruder residence time strengthened microcompartments (Fig. 6b,c) but weakens A/B compartments[45,51,52]. Thus, although compartmentalization remains poorly understood and much more work is required, our results suggest that microcompartments are at least partially distinct from large A compartments.

While our regions are rich in genes and CREs and more work is required to establish generality, our data nevertheless suggest that many CRE interactions (E–P, P–P and E–E) are intrinsically broadly promiscuous and exhibit only moderate intrinsic selectivity. Indeed, we observed dozens of enhancers and promoters that formed >40 distinct loops (Fig. 2f), consistent with our prior work in interphase[28,40]. This notion is further supported by Schooley et al. (cosubmitted with our preprint), who also observed promiscuous microcompartmentalization of CREs in telophase and in nuclear-transport-deficient G1 cells[92]. Moreover, our polymer model assumes no CRE selectivity and that all CREs have equal affinity for each other but nonetheless reproduces experimentally observed microcompartmentalization. As cells exit mitosis into interphase, mechanisms that reduce the promiscuity of CRE interactions include chromosome decompaction upon G1 entry, the constraining and pruning actions of CTCF/cohesin and perhaps the action of potentially selective CRE looping factors such as YY1 and LDB1 (refs. 62,75,92,93), among other mechanisms.

Our observation of transiently peaking microcompartments may explain the hyperactive transcriptional state that forms during mitotic exit, during which about half of all genes transiently spike[1,42–44]. Although our observation that promiscuous CRE interactions lead to microcompartment formation does not mean that all or some CRE interactions are causally instructive for transcription, we nevertheless observed that CRE interactions slightly precede RNA Pol II promoter binding on average (Fig. 3e). Furthermore, we found that genes known to transcriptionally spike during mitosis form more loops (Fig. 4c), longer loops (Supplementary Fig. 5f) and stronger loops (Fig. 4d) compared to nonspiking genes, with a more pronounced transient spike in strength in AT (Fig. 4d,e). This result suggests that microcompartments have a role in facilitating gene activation during mitosis. While correlative in nature, our analyses are consistent with a model in which promiscuous microcompartmentalization of CREs in mitosis occurs first and then subsequently induces the mitotic transcriptional spiking phenomenon, which was observed previously[1,42–44] but lacked an explanation. Altogether, our data and simulations suggest that CREs are intrinsically broadly interaction compatible, leading to microphase-separation-mediated microcompartment formation that peaks in AT, possibly explaining the broad and transient transcriptional spiking observed during mitotic exit[42].

## Online content

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

## Methods

### Experimental procedures

**Overview of the RCMC experiment.** RCMC[28] was developed by merging a modified Micro-C[30] protocol with tiling region capture of a locus[94,95]. The detailed RCMC protocol is provided as Supplementary Information in the original RCMC publication[28]. Here, we summarize the protocol descriptions previously described in Goel et al.[28] in the Supplementary Methods.

The data generated in this paper come from merging of multiple replicates. For the M-to-G1 cell-cycle-synchronized datasets, two RCMC biological replicates were generated for each of the six tested conditions: PM, AT, EG1, MG1, LG1 and the asynchronous condition (Extended Data Fig. 1d). For the condensin degron datasets, four RCMC biological replicates were generated across five tested conditions (0, 0.5, 1, 4 and 8 h of depletion) (Supplementary Fig. 9a–d). Biological replicates were generated by harvesting (culturing, crosslinking, aliquoting and snap-freezing) 1–50 million cells for each tested condition, after which downstream RCMC steps were applied to snap-frozen cell aliquots totaling 15–25 million input cells to generate each biological replicate.

**Cell culture and maintenance.** The G1E-ER4 murine erythroblast cell line was originally gifted by M. Weiss[96]. Two G1E-ER4 sublines were used in this study: G1E-ER4-mCherry–MD[18], and SMC2-AID–mCherry[17]. G1E-ER4-mCherry–MD cells express the mitotic degradation domain of cyclin B fused to mCherry (used for isolating specific cell populations during the M-to-G1 transition)[18]. The SMC2-AID–mCherry subline harbors a homozygous, in-frame insertion (auxin-inducible degron sequence and mCherry) at the *SMC2* locus[17]. All lines were maintained under previously described conditions for G1E-ER4 cells[96].

**Synchronization and isolation of mitotic cell populations.** Synchronization and isolation of cell populations during mitotic exit were carried out as previously described[18] with minor modifications (Fig. 1a and Extended Data Fig. 1a). Briefly, actively dividing cells (density: 0.5–0.8 million cells per ml) were treated with 200 ng ml⁻¹ nocodazole for 8.5 h for PM arrest. Cells were washed with nocodazole-free medium and released for the following time points to enrich for specific populations during the M-to-G1 transition: 40 min (AT), 1 h (EG1), 2 h (MG1) or 4 h (LG1). After nocodazole treatment and release, cells were sequentially crosslinked with formaldehyde and DSG. Cells were centrifuged for 5 min at 1,500 rpm and resuspended in PBS with 1% formaldehyde (1 million cells per ml) and incubated with gentle rocking at room temperature for 10 min. Formaldehyde crosslinking was quenched with 0.375 M Tris pH 7.5 (room temperature, 5 min). Cells were washed twice with cold PBS and resuspended in 3 mM DSG (ProteoChem, c1104-1gm). DSG crosslinking was carried out for 45 min at room temperature with gentle rocking. DSG crosslinking was quenched with 0.375 M Tris pH 7.5 (room temperature, 5 min). Cells were then permeabilized with 0.1% TritonX-100 and stained with mitosis specific anti-pMPM2 antibody (Millipore, 05-368) for 50 min at room temperature (0.5 μl per 10 million cells, 300× dilution). Secondary antibody staining was carried out for 30 min at room temperature with APC-conjugated F(ab′)2-goat anti-mouse (Thermo Fisher Scientific, 17-4010-82) (2 μl per 10 million cells: 80× dilution). Finally, cells were resuspended in FACS buffer with 25 ng ml⁻¹ DAPI and kept on ice. To enrich for populations at specific cell-cycle stages during the M-to-G1 transition, cells were subjected to flow cytometry sorting using the MoFlo Astrios EQ sorter (Beckman Coulter). The following markers were used to isolate specific cell populations: PM, high mCherry–MD, positive pMPM2 and 4 N DAPI; AT, low mCherry–MD and 4 N DAPI; G1 populations, negative mCherry–MD and 2 N DAPI. Sorted cells were aliquoted and flash-frozen in liquid nitrogen.

**SMC2 degradation.** Mitotic depletion of SMC2 was carried out as previously described[17] (Fig. 5a). Briefly, G1E-ER4-SMC2-AID–mCherry cells were first arrested in PM with (200 ng ml⁻¹) nocodazole treatment for 12 or 15 h (for 8-h auxin time point). Toward the end of nocodazole treatment, cells were also treated with auxin (1 mM) to deplete SMC2 for the following time points: 0, 0.5, 1, 4 and 8 h. Total nocodazole treatment time was 12 h for all samples except for the 8-h auxin treatment, which had a total nocodazole treatment time of 15 h. Cells were serially crosslinked as described above with 1% formaldehyde and subsequently 3 mM DSG. Cells were permeabilized and stained with pMPM2 primary antibody and APC-conjugated F(ab′)2-goat anti-mouse (Thermo Fisher Scientific, 17-4010-82) as described above. Cells were subjected to flow cytometry to enrich for PM-arrested samples. All samples were sorted for pMPM2⁺ cells; auxin-treated cells were sorted on the basis of a low mCherry signal (indicative of SMC2 degradation).

### Data analysis

**Mapping and normalizing RCMC.** RCMC paired-end reads generated by the Illumina NovaSeq sequencers were downloaded as FASTQ files for each sample, pair mate and flow cell lane. Read quality was verified using FastQC (version 0.11.9). Paired-end reads were aligned to the UCSC mm39 genome using bwa-mem2 (version 2.2.1). Aligned paired-end reads were then parsed with pairtools (version 0.3.0) using '--add-columns mapq --walks-policy mask --min-mapq 2'. Parsed reads were filtered for PCR duplicates and unmapped or multimapped reads with pairtools dedup using '--max-mismatch 1'. The remaining reads were indexed (pairix version 0.3.7) and filtered (pairtools select) to retain only those reads where both read mates were in a locus of interest (Extended Data Fig. 1b,c and Supplementary Figs. 9c and 10b). These filtered reads were subsequently converted to cool format using cooler (version 0.8.11) cload pairs, creating binned read counts across the genome for 50-bp bins. Finally, cool files were converted to the mcool format with cooler zoomify including the '--balance' option, compiling read counts for bins from 50 bp up to 10 Mb in size.

Contact matrices were balanced using iterative correction and eigendecomposition (ICE)[97] as previously described[28], which normalizes all rows and columns of a contact matrix sum to the same value. ICE balancing was performed on mcool files containing data only within captured ROIs.

**Visualizing RCMC.** RCMC contact maps were visualized alongside genomic annotations and published ChIP-seq datasets using the HiGlass[98] browser (http://higlass.io/) and software (version 0.8.0). Contact maps shown in figures were generated using cooltools[99] (version 0.5.0) (https://cooltools.readthedocs.io/) (Figs. 1a, 2a, 3a and 5b,e, Extended Data Figs. 2d, 4–7 and 8a and Supplementary Figs. 1 and 6–8). Genomic tracks (that is, ChIP-seq) and gene annotations for figures in this paper were generated using CoolBox[100] (version 0.3.3). In generating our genomic tracks, we analyzed 23 public datasets (Supplementary Table 1) using processed bigWig files that were CrossMapped[101] (version 0.6.1) (http://crossmap.sourceforge.net/) to the mm39 reference genome. Tracks were visualized using the Integrative Genomics Viewer (IGV)[102] (version 2.10.3) to scale tracks by identifying local maxima and minimizing noise.

**Comparing data across methods.** Mapped sequencing reads were filtered using pairtools select to quantify read counts according to chosen evaluation criteria (Fig. 1d, Extended Data Figs. 1c and 2b,c and Supplementary Figs. 9c and 10b). Filtering was performed identically across the RCMC and Hi-C datasets on pairs files containing mm39-mapped reads. RCMC pairs files were generated as described above, while mm9-aligned pairs files and loop calls containing all unique reads were downloaded for Hi-C (GSE129997) and CrossMapped to the mm39 genome.

Quantifications of read coverage across bins were calculated in Python using cooler to load unbalanced 250-bp resolution cool files

into memory as matrices. These matrices were then iterated through to determine the average number of interactions in each contact bin (Fig. 1d) and the fractions of bins containing at least one read at different contact distances (Extended Data Figs. 2b,c). Visualizations of read coverage were plotted using cooltools[99] for RCMC and Hi-C data at 5-kb resolution (Extended Data Fig. 3).

Genome-wide equivalents for RCMC data were calculated by extrapolating the number of unique contacts mapped to a capture locus to a region the size of the entire mouse genome. This approach assumes homogeneous read coverage throughout the genome; in reality, however, read coverage is unevenly distributed between regions depending on the specific region and which 3C method is used. Specifically, genome-wide Hi-C also had higher coverage at the *Ctd1* region than the genome-wide average. As such, compared to Hi-C at *Ctd1*, RCMC captured ~351-fold more unique contacts in the most deeply sequenced condition (LG1).

**Replicate reproducibility analysis.** The reproducibility of RCMC replicates (Extended Data Fig. 1e and Supplementary Fig. 10a) was evaluated using HiCRep[103] (version 1.12.2) for contact maps at 5-kb resolution, with parameters lbr = 0 and ubr = 5,000,000. Reproducibility scores were calculated across regions for all replicates using the optimal *h* value determined from a single replicate.

**Contact decaying curve analysis.** Contact decay curves were generated by plotting contact probability against genomic separation using cooltools[99] (Figs. 1c and 5g and Extended Data Fig. 2a). Balanced and smoothed curves were generated using contact matrices across each chromosome binned to 150-bp resolution. RCMC curves were truncated at 1-Mb genomic separation because of noise at larger genomic separations (all loci 1–2 Mb in size).

**Downsampling.** RCMC datasets were downsampled using pairtools sample to randomly select a subset of the mapped contact pairs. To downsample RCMC (Extended Data Fig. 2c,d), a pairs file containing all mapped reads across all replicates was downsampled using downsampling ratios corresponding to ten orders of two from 1:2 to 1:1,024. Each downsampled pairs output file was then filtered for reads with both mates within one of the five captured loci, the unique reads were extracted and an mcool file was generated for visualization and analysis.

**Chromatin loop analysis.** Chromatin loops were initially called on M-to-G1 RCMC data using Mustache[104] (version 1.2.4) (https://github.com/ay-lab/mustache) at data resolutions of 0.25, 0.5, 1, 2, 5 and 10 kb, with sparsity thresholds of 0.7 and *q*-value thresholds of 0.1. Finer resolutions of loop calling identified more microcompartmental loops but still missed many loops while also increasingly misidentifying stripes as loops, overlapping loops and clustering calls at short genomic distances off of the diagonal.

Manual loop calling was subsequently performed in an attempt to minimize these artifacts in microcompartment analysis. We defined loops as punctate foci of interaction (that is, dots), visibly discernible as being enriched relative to their local background. We did not identify diffuse and overly faint interactions, homogeneously enriched stripes and very-short-range loops just off of the diagonal (that is, under ~5 kb of genomic separation) as loops. Loops were called on ICE-balanced M-to-G1 datasets to create a superset of interactions spanning PM through LG1, with the AT and LG1 conditions serving as the primary datasets for manual annotation. Calling was done across data resolutions from 150-bp to 3.2-kb resolution using the HiGlass browser interface. Scale bar limits were dynamically modulated to minimize background and clearly distinguish focal enrichment. A total of 3,350 loops spanning 363 loop anchors were manually annotated across the five loci (Fig. 2d,e).

Previously published M-to-G1 Hi-C loop calls (GSE129997) were downloaded and lifted over to mm39-aligned coordinates (Extended Data Fig. 8a). Calls within the capture loci were merged across the five M-to-G1 conditions, with calls within 10 kb of one another merged into a single loop call with averaged coordinates to avoid redundancy. A total of 134 loops spanning 227 unique anchors were found across the five loci (Extended Data Fig. 7d,e).

**Loop anchor classification using inclusive and exclusive approaches.** To classify loop anchors as promoter, enhancer or bound to CTCF and cohesin (Fig. 2d and Extended Data Fig. 8d), loop anchor locations were compared to the corresponding chromatin features as follows. Promoter regions were defined using all TSS locations in the mm39 UCSC RefGene annotation[105] ± 2 kb. Enhancers or CTCF-bound and cohesin-bound sites were defined on the basis of overlap of H3K4me1 (GSM946535) and H3K27ac (GSE61349) or CTCF (GSE129997) and RAD21 (GSE129997), respectively. For all datasets, bigWig files were converted to bedgraph files using UCSC bigWigToBedGraph (version 377)[106], followed by peak calling using MACS2 bdgpeakcall[107] (version 2.2.7.1). For CTCF, called peaks were then overlapped with CTCF sites identified using FIMO (version 5.4.1). First, fasta-get-markov was used to generate a background model using the mm39 genome assembly. Then, motifs were identified using '--max-stored-scores 50000000 --thresh 1e-3'. Finally, locations of motifs were overlapped with peaks identified in ChIP-seq data and only the motif with the highest score for each peak was maintained. The peaks (H3K4me1, H3K27ac and RAD21) or identified sites (CTCF) were then overlapped using BEDTools intersect (version 2.30.0) to give enhancers or CTCF-bound and cohesin-bound regions. Anchors of interactions ± 1 kb were then overlapped with each of the three features to classify them as promoter, enhancer or bound to CTCF and cohesin. Anchors overlapping none of these three features were classified as 'other'.

In some case, the anchor fit multiple categories, for example, in cases where a CRE (enhancer or promoter) was also bound by CTCF and cohesin. To accommodate these cases, we used an inclusive or exclusive classification depending on the analysis. For analyses where we used the inclusive classification, we took a hierarchical approach classifying all promoters as promoters even if they overlapped other features and then classifying all enhancers as enhancers even if they overlapped other features, leaving CTCF and cohesin anchors as those bound by CTCF and cohesin but not overlapping promoters and enhancers. For analyses where we used the exclusive classification, any anchors that overlapped both enhancers or promoters and CTCF or cohesin, as well as any loops formed by these anchors, were removed from consideration, limiting the analysis to 'pure' anchors and loops representative of single rather than multiple organizational mechanisms.

We used the inclusive classification for all contact map loop overlays and the following figures: Figs. 2b–f and 4a–c, Extended Data Fig. 8b–e and Supplementary Fig. 5c–f. We used the exclusive classification for all loop strength calculations to avoid obfuscating the contribution of different organizational mechanisms toward the temporal dynamics of M-to-G1 loop formation. Specifically, the exclusive classification was used for the following figures: Figs. 2g, 3b–e and 5c,d and Supplementary Figs. 3, 4 and 9e,f.

The number of interactions formed by each anchor and the lengths of the interactions they form (Figs. 2b,c,f and 4a, Extended Data Fig. 8b,c and Supplementary Fig. 5f) were determined and visualized in Python using the matplotlib package.

**Heat map and metaplot generation.** Heat maps and metaplots were generated for annotated loop anchors using deepTools[108] (version 3.5.1) computeMatrix followed by plotHeatmap in a region ±1 kb around the center of each anchor for genomics data listed in Supplementary Table 1 (Supplementary Fig. 2).

**Calculation of background-subtracted loop strength in RCMC contact maps.** The loop strength score is calculated as a sum of values of the iteratively corrected Micro-C contact map minus local background within a 21 × 21 square window centered on the loop (Figs. 2g, 3c–e, 4b,c and 5c and Supplementary Figs. 3a, 4 and 9e). At a resolution of 500 bp, this window encompasses interactions whose anchors are within ±5 kb from the loop. The local background matrix, which is also a 21 × 21 square window centered on the loop, is constructed such that each matrix element corresponding to a genomic separation $s$ is the average of all matrix elements of genomic distance $s$ within ±100 kb of the loop. The local background matrix is designed to represent the expected number of distance-dependent random contacts in the absence of an active looping mechanism.

**Pileup and loop strength analysis.** Pileup visualizations and intensity quantifications of annotated looping interactions were performed using cooltools[99] to generate aggregate peak analyses (APAs; Figs. 3b and 5d and Supplementary Figs. 3b,c and 9f) and individual loop strength plots (Figs. 2g, 3c–e, 4b,c and 5c and Extended Data Figs. 3a, 4 and 9e). Plots for all loops of a given classification (for example, E–P loops) were generated and analyzed individually or averaged for a 24-kb (Fig. 3b and Supplementary Figs. 3b,c and 4) or 20-kb (Fig. 5d and Supplementary Fig. 9f) window centered on the loop at 250-bp resolution. Strength was calculated as the observed signal divided by expected signal throughout the paper, with observed signal being calculated as the integral of ICE-balanced matrix values within 5 kb of the loop and expected signal being calculated as the local $P(s)$ curve at the loop. The sole exception to the use of observed over expected strength calculations is Fig. 3d, in which simply the observed (total) signal was considered to directly compare the percentage change between two conditions (AT and LG1) with notably different $P(s)$ curves.

**Compartmentalization analysis.** Compartments were called by applying eigendecomposition to the 0-h, 1-h and 4-h condensin depletion RCMC contact matrices using cooltools[99] (Fig. 5e,f). RCMC data were first ICE-normalized to remove the distance-dependent effect of contact frequency. Eigendecomposition was then performed, with G+C content serving as a correlate for orienting eigenvectors to indicate A (gene-rich or active chromatin) or B (gene-poor or inactive chromatin) compartments. Finally, eigenvectors were binarized and visualized as tracks and map overlays, as well as region-specific compartmentalization saddle plots, using cooltools[99]. Compartment calling and saddle plot generation was performed at data resolutions of 0.5, 1, 2 and 5 kb, with all producing similar output (calls at 2-kb resolution are shown for clarity).

**Identifying transcriptionally spiking genes.** To quantify transcription, we calculated the average signal of Pol II ChIP (GSE129997) within a 2.5-kb window at all five time points for all TSSs in the capture regions. For genes with multiple TSSs, we only took the single dominant TSS (that is, the one with the highest Pol II signal). We then filtered for active genes, which we defined as those where the Pol II signal was at least 0.5 for all time points except PM; this resulted in 88 genes for further analysis. For each gene, we normalized the Pol II signal by dividing each value by the average value in the three G1 time points. We used these normalized Pol II ChIP values to perform PCA (Fig. 5b). Genes with positive values in PC1 were labeled as spiking genes and those with negative values in PC1 were labeled as nonspiking genes.

## Polymer simulations and analysis

**Polymer simulations.** We performed polymer molecular dynamics simulations using custom-written code (https://github.com/mirnylab/microcompartments) that uses the polychrom library[109] (https://github.com/open2c/polychrom/), which is a Python wrapper for the OpenMM molecular simulation toolkit[110,111]. These codes are freely and publicly available. Simulations were conducted by coupling one-dimensional (1D) loop extrusion simulations to 3D polymer simulations. Genomic positions of loop extruders were used to determine which polymer sites are physically bridged at a given instant time. Between loop extrusion simulation steps, the chromatin polymer evolves under the constraints set by the loop extruders bridging polymer sites. Simulation parameters are given in Supplementary Tables 2 and 3.

**Modeling loop extrusion.** We simulate $N$ loop extruders on a 1D lattice of length $L = 61,600$ genomic sites, where each site represents $\sigma = 0.5$ kb of chromatin (parameters in Supplementary Table 2). The mean distance between loop extruders is given by $d = L/N$; thus, the linear density of extruders is $1/d$. In simulation sweeps, $d = 100$ kb unless noted. Each loop extruder occupies (and, in the 3D simulation, bridges) two genomic sites. Each loop extruder is loaded at two adjacent unoccupied sites. Subsequent to loading, each of the two components of the loop extruder may translocate away from the position at which it was loaded with probability $p$, which leads to a macroscopic loop growth velocity $v = 2p\sigma/\tau_0$, where $\tau_0$ is the time between loop extrusion timesteps, taken to be 0.5 s. Following in vitro observations[8,112–115], $v = 1$ kb s$^{-1}$ unless noted. Extrusion by a loop extruder component continues until (1) the loop extruder is stochastically unloaded at rate $1/\tau_{res}$, where $\tau_{res}$ is the mean residence time; (2) the loop extruder component encounters another loop extruder component; or (3) if the loop extruder models cohesin, it encounters a properly oriented CTCF stall site and stalls with probability $q = 0.5$. CTCF sites for the simulated *Dag1* region were annotated from the intersection of CTCF and RAD21 ChIP-seq peaks in the mm39 genome assembly. CTCF sites were only included in M-to-G1 transition simulations. For simulations with quenched (inactive) loops (as in Extended Data Fig. 9g), 1D active extrusion simulations were performed for $10^6$ extrusion steps to allow the distribution of loops to equilibrate. Loop extruders were then frozen in place on the polymer chain and maintained their polymer loops throughout the remainder of the simulation. This protocol was performed for each of at least ten simulations, generating a unique loop configuration for each polymer simulation; given that each polymer contained eight copies of the simulated *Dag1* locus, we simulated ≥80 static loop configurations within *Dag1* for each set of parameters.

**Modeling polymer dynamics.** A chromosome was modeled as a chain of $L = 61,600$ monomeric subunits, each representing $\sigma = 0.5$ kb of chromatin (Supplementary Table 2). Consecutive subunits were connected by harmonic springs. Monomers also interacted through soft repulsive interactions ($E_{repel} = 3\,k_BT$), where modeling excluded volume. Monomers were assigned as one of three types (A, B or C; Fig. 6a) and could interact through attractive interactions, with homotypic affinities of $\epsilon_A = 0$, $\epsilon_B = 0.05\,k_BT$ and $\epsilon_C = 0.9\,k_BT$ and heterotypic affinities of 0 unless noted. As in previous studies[45,116], the interaction potential between monomers a distance $r$ apart was given by a smooth square-like potential:

$$U(r) = \begin{cases} E_{repel}\left(1 + \frac{1}{E_0}\left(\left(\frac{a_0}{a}r\right)^2 - 1\right)\left(\frac{a_0}{a}r\right)^{12}\right) \text{for } r < a \\ -\epsilon_i\left(\frac{1}{E_0}\left(\left(\frac{r-(a+a^*)/2}{(a^*-a)/2}a_0\right)^2 - 1\right)\left(\frac{r-(a+a^*)/2}{(a^*-a)/2}a_0\right)^{12} + 1\right) \text{for } r > a \end{cases}$$

where $a$ is the monomer diameter (described below), $a^* = 1.5a$, $a_0 = \sqrt{6/7}$, $E_0 = 46,656/823,543$ and $i$ denotes the type of homotypic compartmental interaction, if applicable. Assignment of monomer types was based on the analysis and annotation of the 1.925 Mb *Dag1* region, with C-type monomers (microcompartments) assigned on the basis of non-CTCF AT loop calls. C-type regions were uniformly taken to be 1.5 kb long. In each chromosome, eight 3,850-monomer *Dag1* regions were simulated, with 3,850 neutral (A-type) monomer segments intercalating between *Dag1* regions. Simulation results

were insensitive to whether isolated *Dag1* regions or *Dag1* repeats were simulated. In simulation sweeps, simulated chromosomes were confined to a spherical region at the prescribed (volumetric) density $\rho_{chr} = 0.25$ unless otherwise noted. Simulations were evolved using the OpenMM fixed timestep Langevin integrator with (polychrom variables) timestep = 40 and collision_rate = 0.01. Polymer simulations were evolved for 540 time steps between loop extrusion update steps (time calibration described below), except where noted (Extended Data Fig. 10e).

**Equilibration, time calibration and data collection.** Equilibrium simulations (for parameter sweeps) were equilibrated by first evolving 1D loop extruder dynamics for $10^6$ extrusion time steps ($\gg \tau_{res}$). The 3D chromosomes with active loop extrusion were then equilibrated by simulating for $>30 \times 10^6$ polymer time steps ($>20\tau_{res}$). Equilibration was assessed by inspection of contact frequency curves and contact maps. Data were then collected from 1,500 time points over $27 \times 10^6$ polymer time steps. Data were collected for a minimum of ten independently equilibrated simulations per condition (that is, ≥80 copies of the simulated *Dag1* region). RCMC-like contact maps were generated using a contact radius of four monomers and bin size of four monomers (2 kb).

Polymer and loop extruder simulation times were calibrated to each other and experimental timescales by measuring the two-point mean-squared displacement and root-mean-squared radius of gyration of a 515-kb region in simulations without loop extrusion. These measurements were compared to experimental measurements of the *Fbn2* locus in interphase mouse embryonic stem cells[55] under cohesin degradation conditions (4 h RAD21–mAID depletion); this resulted in a monomer size of $a = 25$ nm. Together with the choice of loop growth speed of 1 kb s$^{-1}$, this procedure resulted in the selection of 540 polymer steps per loop extruder step, with each loop extruder time step equivalent to 0.5 s.

**Modeling the M-to-G1 transition.** Simulations of the M-to-G1 transition were performed by first equilibrating loop extruder dynamics for 36,000 extrusion steps (~5 $\tau_{res}$ for simulated condensin II) and then performing full polymer simulations with extrusion in cylindrical confinement (4:1 aspect ratio) at density $\rho_{chr} = 0.65$ for 300 min (that is, 36,000 more extrusion steps with 540 polymer steps per extrusion step). Ends of the chromosome were tethered to opposite ends of the cylinder. Simulations proceeded as described in the text, with data collection for PM at times $0 < t < 10$ min, AT at $20 < t < 30$ min, EG1 at $55 < t < 65$ min, MG1 at $115 < t < 125$ min and LG1 at $235 < t < 245$ min.

During PM, for $t < 15$ min, loop extruders were condensins I and II with respective residence times $\tau_{res}^{CI} = 3$ min and $\tau_{res}^{CII} = 1$ h and mean separations $d^{CI} = 35$ kb and $d^{CII} = 140$ kb (unless otherwise noted). Over the time $15 < t < 17$ min, the condensin I level transiently increased to $d^{CI} = 27$ kb before gradually decreasing to zero over $17 < t < 20$ min.

During AT, over the time $25 < t < 30$ min, the cylinder height decreased by half, while the radius increased in a way that gradually decreased chromatin density to $\rho_{chr} = 0.45$. Over $30 < t < 35$ min, the strength of the cylindrical confining potential was gradually decreased to zero while a new spherical confining potential with radius corresponding to $\rho_{chr} = 0.25$ was gradually strengthened from zero.

Additionally, at $t = 30$ min, condensin II was removed, A/B compartment interactions were activated and CTCFs were added to the simulation. At this time, we began loading cohesin loop extruders with $\tau_{res}^{cohesin} = 10$ min to the simulation. Cohesin loading continued throughout the simulation of G1 such that the cohesin level increased linearly with time and peaked at the end of the simulation with mean separation $d^{cohesin} = 100$ kb.

Supplementary Table 3 provides a list of parameters used in the model.

**Computation of loop strengths.** Loop strengths in simulations were computed similarly to as described in experiments. Each loop strength (Fig. 6f, middle) was computed as the mean number of contacts within a 6-kb × 6-kb window centered on the microcompartment or CTCF site of interest. To obtain relative loop strengths (Fig. 6e, left), individual loop strengths were scaled by the average number of contacts at the corresponding genomic distance (results were minimally quantitatively altered by scaling by the local mean number of contacts instead).

### Statistics and reproducibility
For RCMC experiments, no statistical method was used to predetermine sample size. For RCMC of untreated cells across the M-to-G1 transition, two biological replicates were generated for each time point and the asynchronous condition. For the condensin degron RCMC experiments, four biological replicates were generated for each depletion duration. Reproducibility across replicates was verified using HiCRep[103] (version 1.12.2). In all statistical analyses, full distributions and individual data points were plotted whenever possible. Loop strength and RNA Pol II signal across the M-to-G1 transition, shown as averages across loops or genes in Fig. 3c,e, were further analyzed on an individual basis in Fig. 4. In Fig. 4, error bars denote the s.e.m. No replicates were excluded from analysis and no data were excluded outside of the filtering steps described above. Randomization and blinding were not applicable because our study did not involve selecting treatment groups from a population.

### Reporting summary
Further information on research design is available in the Nature Portfolio Reporting Summary linked to this article.

## Data availability
Sequencing data are available from the National Center for Biotechnology Information Gene Expression Omnibus under accession number GSE276657.

## Code availability
RCMC analysis code (https://github.com/ahansenlab/RCMC_mitosis_analysis_code) and polymer simulation code (https://github.com/mirnylab/microcompartments) are available from GitHub.

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

## Acknowledgements

We thank B. Zhang, J. Paggi, E. Apostolou, J. Kæstel-Hansen, M. Nagano, C. Hong, S. Nemsick, M. Huseyin, M. Gabriele, S. Kim and the A.S.H. lab for feedback on the paper. We also thank E. Navarrete and the other members of the L.A.M. lab for helpful discussions. A.S.H. acknowledges funding support from the National Institutes of Health (NIH; DP2GM140938, R33CA257878, R01EB035127, UM1HG011536, R01CA300848 and R03OD038390), a National Science Foundation (NSF) CAREER award (2337728), the Gene Regulation Observatory of the Broad Institute of MIT and Harvard, the Pew-Stewart Scholar for Cancer Research, the Mathers Foundation and an RSC award from the MIT Westaway Fund. This work was supported by the Bridge Project, a partnership between the Koch Institute for Integrative Cancer Research at MIT and the Dana-Farber Harvard Cancer Center. We thank the MIT Koch Institute's Robert A. Swanson (1969) Biotechnology Center for technical support, specifically the Integrated Genomics and Bioinformatics Core and MIT BioMicroCenter. This work was supported in part by the Koch Institute Support (core) grant P30-CA14051 from the National Cancer Institute. L.A.M. and E.J.B. are supported by the NIH National Institute of General Medical Sciences GM114190 and NSF awards 2044895 and 2210558. L.A.M. is a Simons Investigator of the Simons Foundation. G.A.B. is supported by 5U01DK127405-04. N.G.A. is supported by 1F31DK136200-01A1. We also thank the Walk-Up Sequencing services of the Broad Institute of MIT and Harvard. The funders had no role in study design, data collection and analysis, decision to publish or preparation of the paper.

## Author contributions

V.Y.G., N.G.A., L.A.M., G.A.B., E.J.B. and A.S.H. designed the research. V.Y.G., N.G.A. and L.P.M. performed the experiments. V.Y.G. and J.M.J. analyzed the experimental data. E.J.B. performed and analyzed the simulations. H.Z. contributed key reagents and expertise. V.Y.G., E.J.B. and A.S.H. drafted the paper with input from L.A.M. and G.A.B. All authors discussed the results and commented on the paper. A.S.H. supervised the study.

## Competing interests

V.Y.G. and A.S.H. previously filed a patent application for RCMC. The other authors declare no competing interests.

## Additional information

**Extended data** is available for this paper at https://doi.org/10.1038/s41594-025-01687-2.

**Correspondence and requests for materials** should be addressed to Edward J. Banigan or Anders S. Hansen.

**a**                    Overview of M-to-G1 sample cell collection by flow sorting

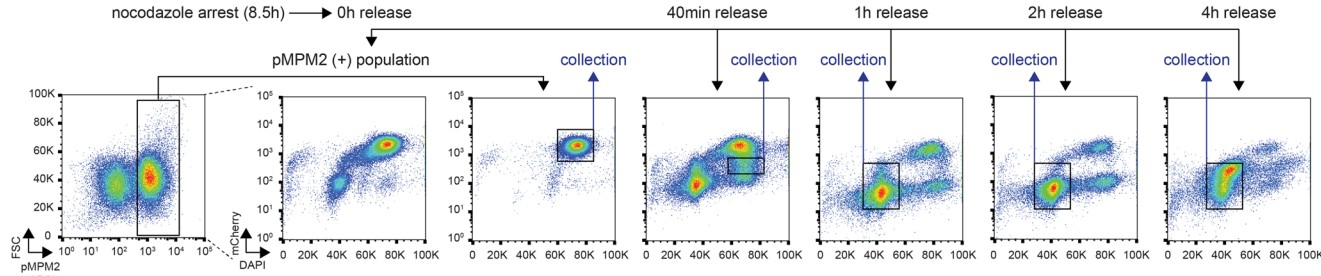

**b**        RCMC capture efficiency by locus, asynchronous condition

| RCMC locus | | *Id1* | *Klf1* | *Cdt1* | *Dag1* | *Myc* |
|---|---|---|---|---|---|---|
| Chromosome | | chr2 | chr8 | chr8 | chr9 | chr15 |
| Coordinates *(mm39)* | | 151,920,000–153,000,000 | 84,846,629–85,856,629 | 122,950,000–124,110,000 | 106,675,000–108,600,000 | 61,810,000–63,684,000 |
| Locus size (Mb) | | 1.080 | 1.010 | 1.160 | 1.925 | 1.874 |
| Biological Rep 1 | Mapped reads *(combined capture)* | | | 546,984,849 | | |
| | Mapped reads *(to locus)* | 50,804,677 | 50,915,842 | 67,166,399 | 82,910,072 | 47,419,514 |
| | Capture efficiency | | | 54.7% | | |
| Biological Rep 2 | Mapped reads *(combined capture)* | | | 448,572,694 | | |
| | Mapped reads *(to locus)* | 30,580,165 | 31,836,300 | 40,547,834 | 49,554,868 | 28,707,523 |
| | Capture efficiency | | | 40.4% | | |

**c**        Comparison of M-to-G1 data density across RCMC and Hi-C

| Read counts comparison | Locus size (Mb) | Unique reads genome-wide | Unique reads entirely in ROIs |
|---|---|---|---|
| RCMC, asynchronous (async) | N/A | 682,289,559 | 246,293,481 |
| *RCMC, Id1 locus (chr2)* | 1.080 | N/A | 40,123,900 |
| *RCMC, Klf1 locus (chr8)* | 1.010 | N/A | 41,737,270 |
| *RCMC, Cdt1 locus (chr8)* | 1.160 | N/A | 55,698,589 |
| *RCMC, Dag1 locus (chr9)* | 1.925 | N/A | 69,310,808 |
| *RCMC, Myc locus (chr15)* | 1.874 | N/A | 39,422,914 |
| RCMC, prometaphase (PM) | N/A | 586,610,326 | 137,330,275 |
| RCMC, ana/telophase (AT) | N/A | 580,899,254 | 171,485,290 |
| RCMC, early G1 (EG1) | N/A | 684,871,630 | 233,015,417 |
| RCMC, mid G1 (MG1) | N/A | 642,882,309 | 229,120,128 |
| RCMC, late G1 (LG1) | N/A | 754,484,446 | 315,514,810 |
| Hi-C, Zhang et al. (2019), PM | N/A | 379,448,977 | 569,540 |
| Hi-C, Zhang et al. (2019), AT | N/A | 409,341,246 | 697,125 |
| Hi-C, Zhang et al. (2019), EG1 | N/A | 400,670,452 | 975,455 |
| Hi-C, Zhang et al. (2019), MG1 | N/A | 362,122,076 | 924,057 |
| Hi-C, Zhang et al. (2019), LG1 | N/A | 445,457,382 | 1,227,761 |

**d**        RCMC cell inputs by condition & replicate

| | Replicate 1 | Replicate 2 |
|---|---|---|
| asynchronous | 10M (across 2 tubes) | 10M (across 2 tubes) |
| prometaphase (PM) | 20M (across 4 tubes) | 16M (across 3 tubes) |
| ana/telophase (AT) | 14.5M (across 3 tubes) | 11.5M (across 2 tubes) |
| early G1 (EG1) | 25M (across 5 tubes) | 25M (across 5 tubes) |
| mid G1 (MG1) | 25M (across 5 tubes) | 25M (across 5 tubes) |
| late G1 (LG1) | 25M (across 5 tubes) | 25M (across 5 tubes) |

**e**        Reproducibility analysis of M-to-G1 RCMC across conditions and replicates

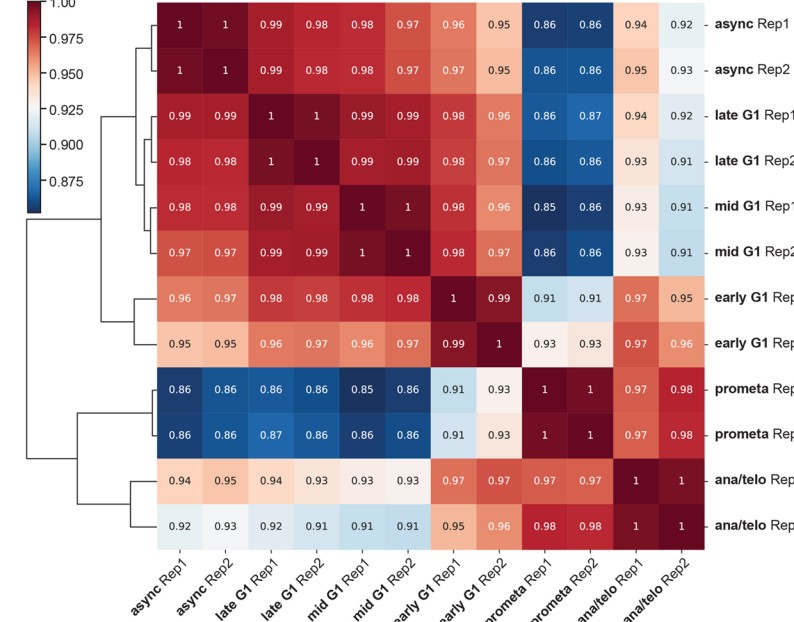

**Extended Data Fig. 1 | See next page for caption.**

**Extended Data Fig. 1 | Overview of RCMC data collection, depth, and reproducibility. a**) Experimental workflow for nocodazole arrest-release cell sample collection for RCMC. Flow cytometry plots show the representative gating strategy used to select pure M-to-G1 cell populations based on pMPM2 (prometaphase), mCherry (MD), and DAPI (DNA) signal. (**b**) Summary of the capture efficiency for each of the five regions for which probes were designed. The locations and sizes of the regions, the number of total mapped fragments genome-wide, the number of mapped fragments with at least end within a captured region, and the capture efficiencies are given. (**c**) Table of uniquely mapped RCMC reads genome-wide and within captured ROIs for the five M-to-G1 transition timepoints and the asynchronous conditions.

Locus-specific quantifications of unique contacts are shown for the asynchronous condition and reflect ligation products for which both fragment ends are within the captured ROI. A comparison against previously published Hi-C data[18] is also provided. (**d**) Table of M-to-G1 cell sample inputs for RCMC, separated by condition and replicate. Samples were split across multiple tubes treated in parallel containing ~5 M cells apiece from MNase digestion onwards and recombined into a single tube during library prep. (**e**) Measurement of reproducibility between M-to-G1 RCMC samples across conditions and replicates. Reproducibility scores are determined using HiCRep at 5 kb resolution, averaged across all five captured loci, and clustered according to similarity.

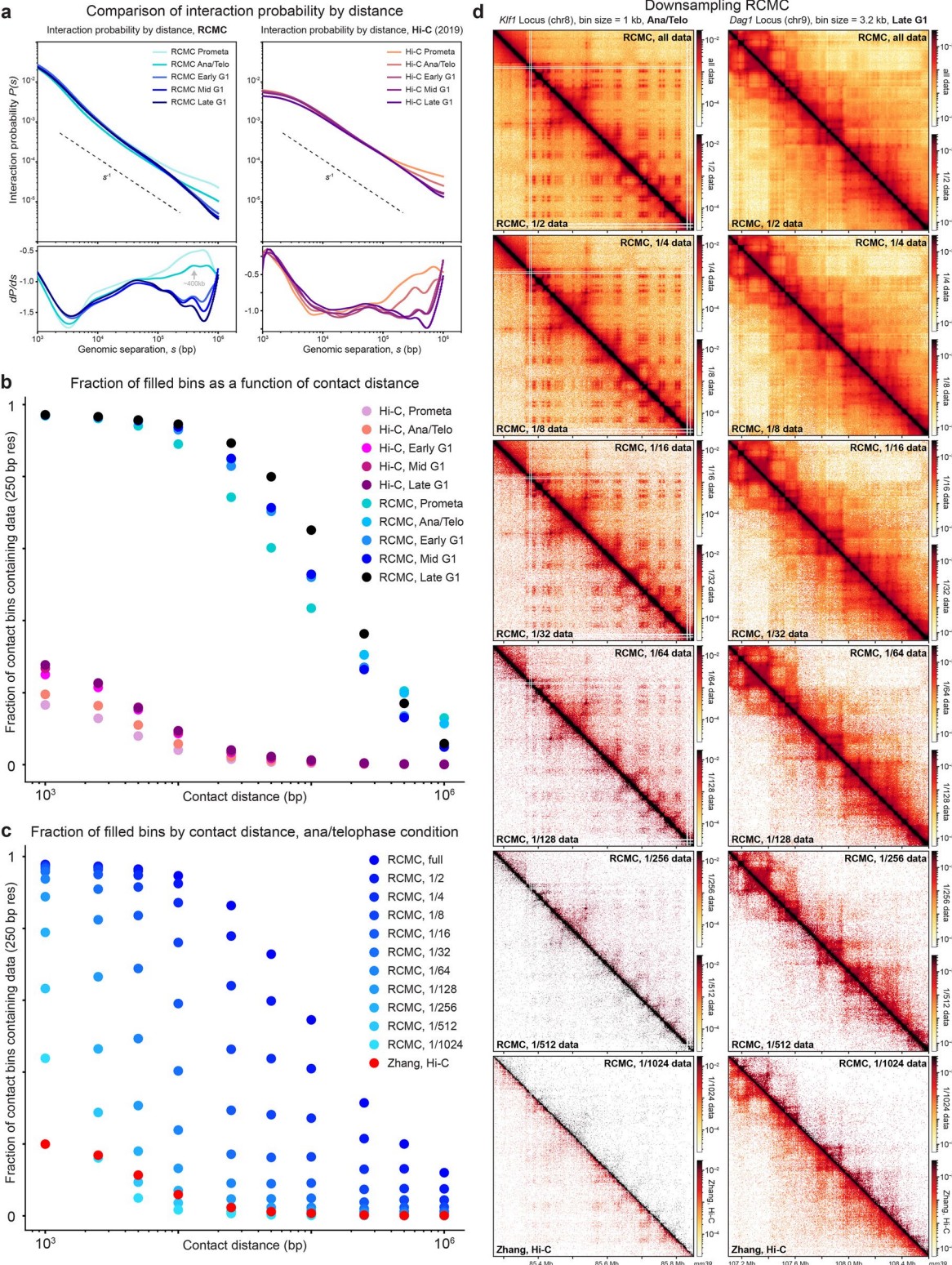

**Extended Data Fig. 2 | RCMC more deeply and efficiently maps target loci than Hi-C.** (**a**) Interaction probability curves comparing the interaction frequency at different genomic separations (*s*) for the five M-to-G1 conditions, shown for RCMC on the left and Hi-C[18] on the right. The first derivative of these *P*(*s*) curves is shown at the bottom. (**b**) Benchmarking comparison of RCMC's ability to fill out high-resolution contact matrices against Hi-C[18]. Region-averaged calculations are shown for both methods across the M-to-G1 datasets. The x-axis shows the contact distance in bp, and the y-axis shows the fraction of all bins at a given contact distance within the captured locus that contain at least one read

at 250 bp resolution. (**c**) As in (**b**), benchmarking comparison of successively downsampled RCMC's ability to fill out high-resolution contact matrices against Hi-C for the ana/telophase condition. Downsampling was applied to all mapped ligated read pairs in increasing powers of 2 until (1/2)[10], or 1/1024[th], of the initial dataset. (**d**) Contact map comparisons of successively downsampled RCMC data, starting from the full dataset (topmost) down to 1/1024[th] (bottom) and with a comparison against the Hi-C dataset. Maps are shown for the *Klf1* locus at 1 kb resolution in the ana/telophase condition on the left and for the *Dag1* locus at 3.2 kb resolution in the late G1 condition on the right.

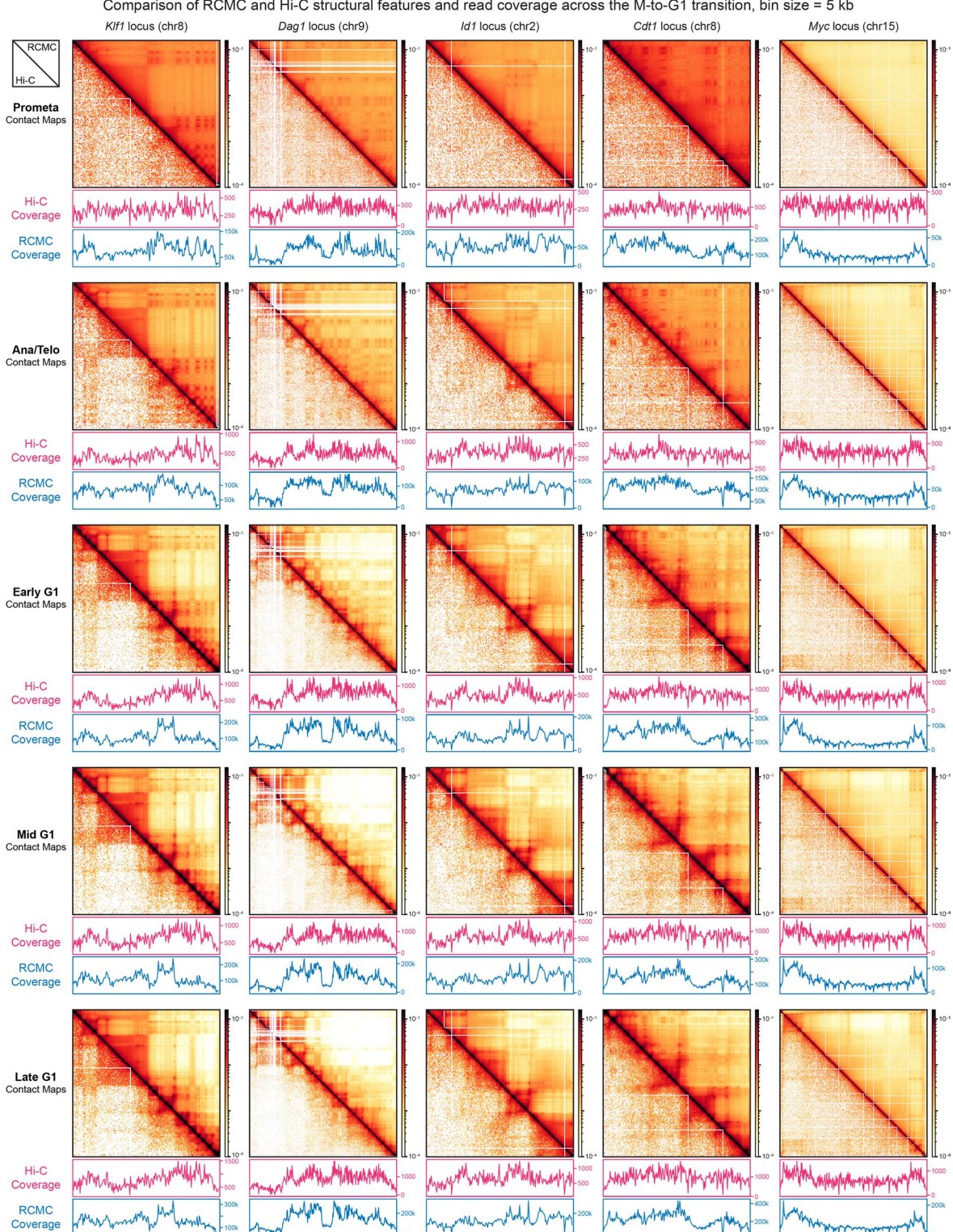

**Extended Data Fig. 3 | Read coverage profiles for Hi-C and RCMC are similar across the M-to-G1 transition.** Whole-locus contact maps for all captured loci across the M-to-G1 transition, shown at 5 kb resolution with RCMC data above the contact diagonal, Hi-C data[18] below the diagonal, and signal intensity scales to the right. Read coverage profiles are shown below the contact maps for each locus and M-to-G1 stage, with Hi-C read coverage in pink and RCMC read coverage in blue.

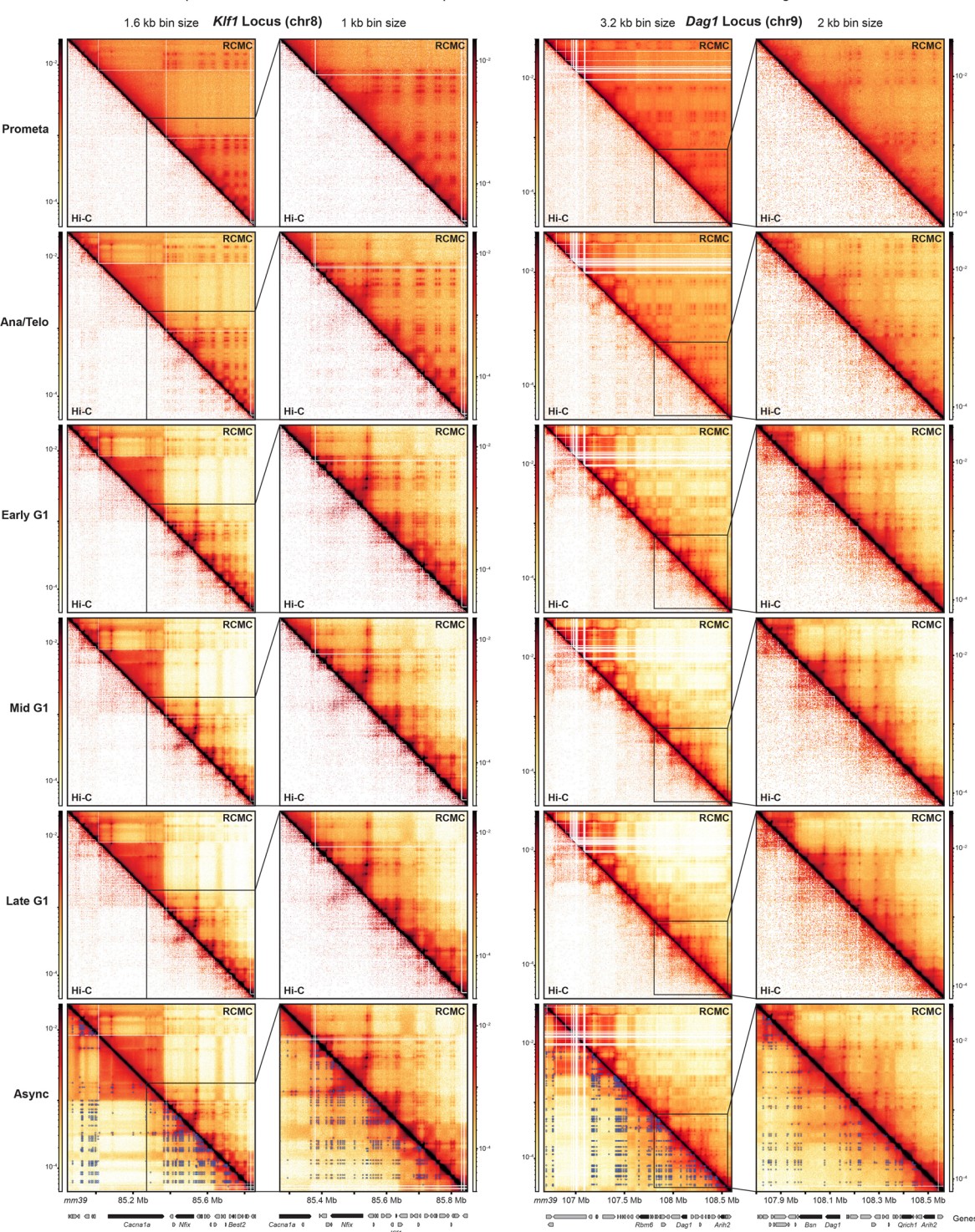

**Extended Data Fig. 4 | RCMC maps mitotic exit at the *Klf1* and *Dag1* loci.**
Contact map comparisons of RCMC against previously generated Hi-C data[18] at the *Klf1* and *Dag1* loci across the M-to-G1 transition. Full capture regions are shown for both loci at 1.6 kb and 3.2 kb resolution, respectively, along with zoom-ins at 1 kb and 2 kb resolution, respectively. The asynchronous RCMC dataset is shown at the bottom, with an overlay of the superset of all annotated interactions below the diagonal. Gene annotations are shown below the contact maps and signal intensity scales are shown next to the maps.

Comparison of RCMC and Hi-C contact maps across the M-to-G1 transition at the *Id1* and *Cdt1* loci

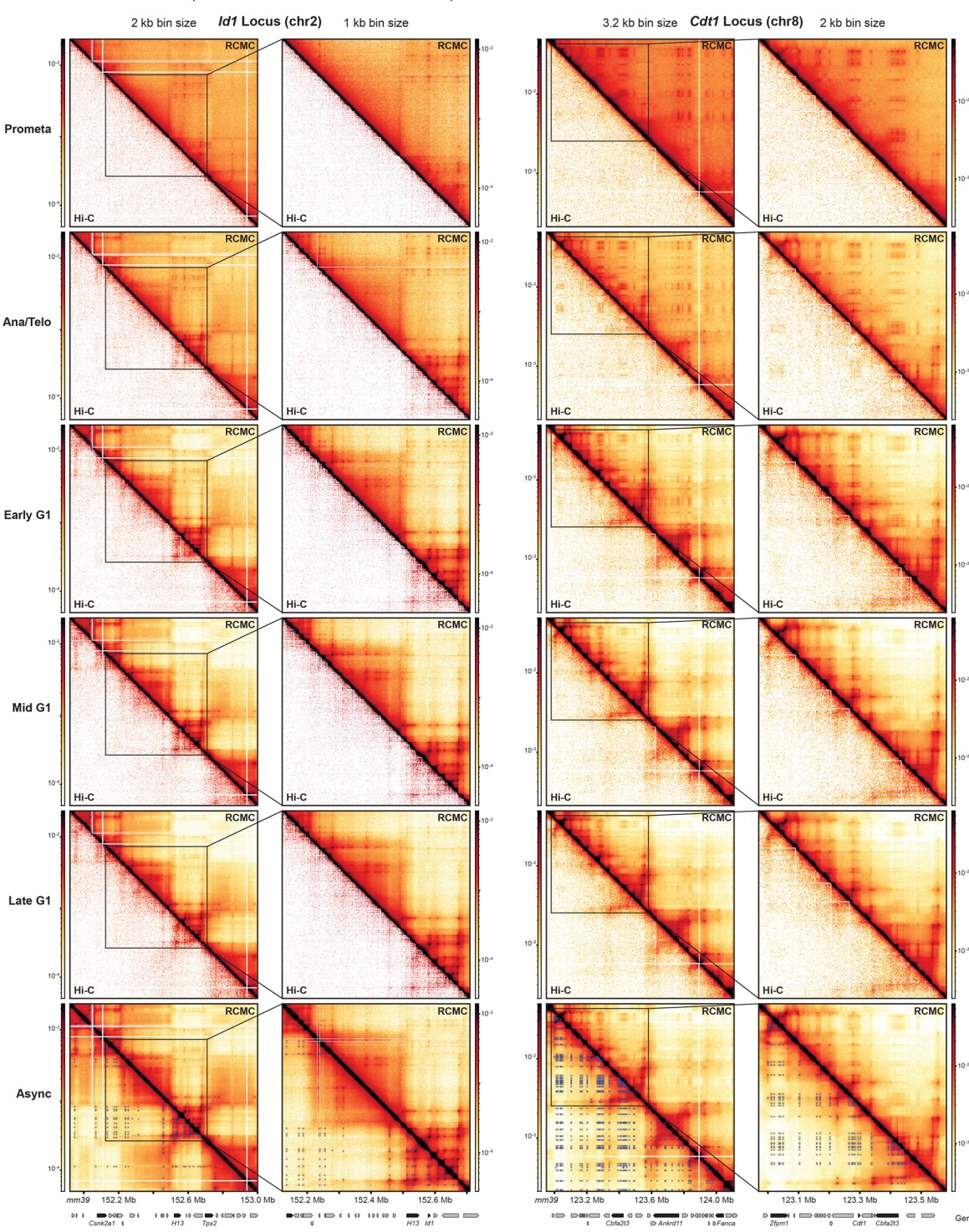

**Extended Data Fig. 5 | RCMC maps mitotic exit at the *Id1* and *Cdt1* loci.** Contact map comparisons of RCMC against previously generated Hi-C data[18] at the *Id1* and *Cdt1* loci across the M-to-G1 transition. Full capture regions are shown for both loci at 2 kb and 3.2 kb resolution, respectively, along with zoom-ins at 1 kb and 2 kb resolution, respectively. The asynchronous RCMC dataset is shown at the bottom, with an overlay of the superset of all annotated interactions below the diagonal. Gene annotations are shown below the contact maps and signal intensity scales are shown next to the maps.

Asynchronous RCMC maps of captured loci with all associated ChIP datasets

**Extended Data Fig. 6 | Finely resolved structures in RCMC can be aligned to 1D genomics data.** Asynchronous RCMC contact maps are shown for the *Klf1*, *Dag1*, *Id1*, and *Cdt1* loci alongside zoom-ins at finer resolutions. The superset of annotated interactions is shown below the diagonal. Gene annotations and ChIP-seq tracks (Supplementary Table 1) are shown below the contact maps, while signal intensity scales are shown next to the maps. ChIP-seq tracks include condition-separated CTCF, RAD21, and RNA Pol2 datasets and asynchronous H3K4me1, H3K4me3, H3K9me3, H3K27me3, and H3K36me3 datasets.

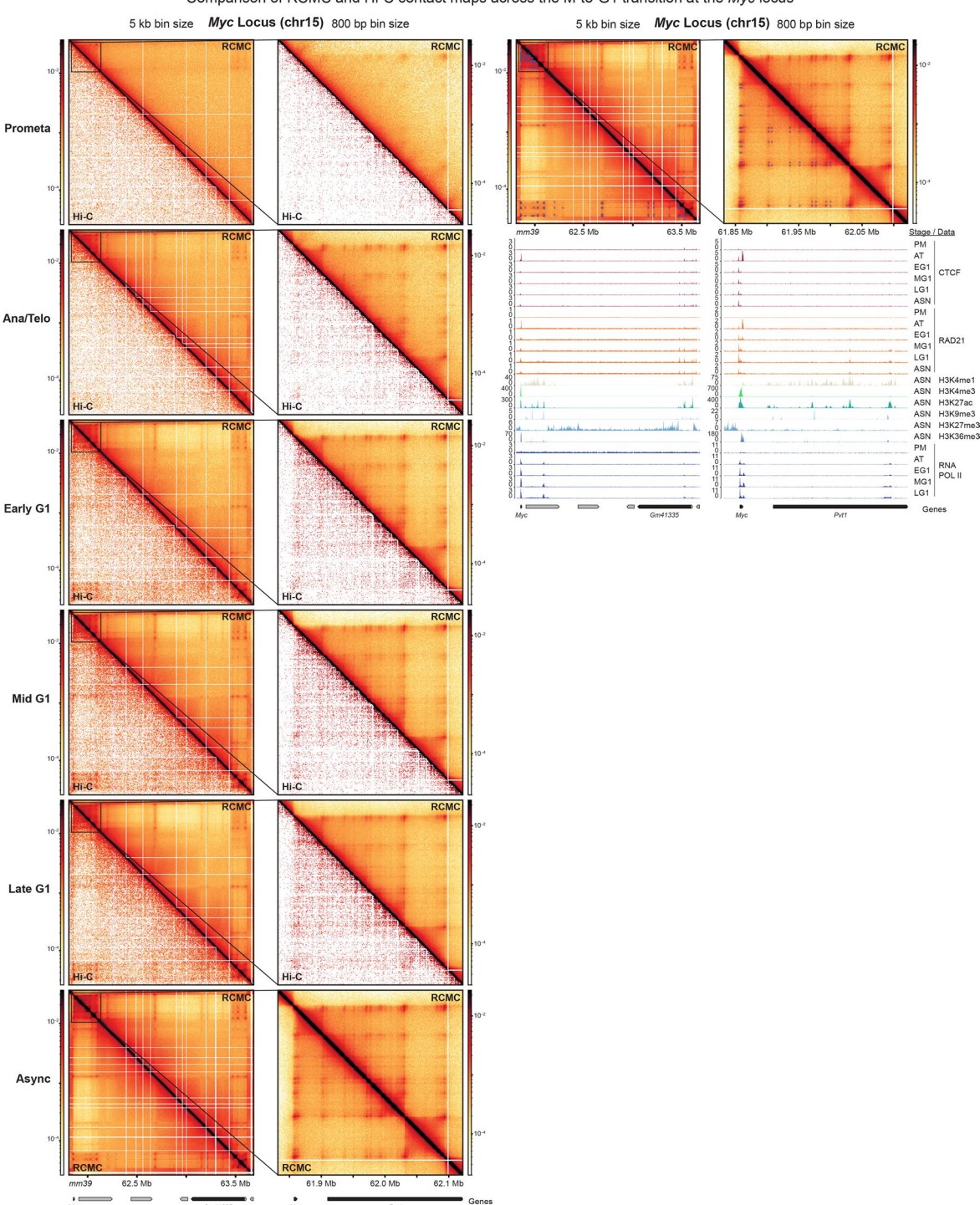

**Extended Data Fig. 7 | RCMC maps mitotic exit at the *Myc* locus.** (Left) Contact map comparisons of RCMC against previously generated Hi-C data[18] at the *Myc* locus across the M-to-G1 transition. The full capture region is shown at 5 kb resolution along with a zoom-in at 800 bp resolution, and the asynchronous RCMC dataset is shown at the bottom. (Right) Contact maps of the asynchronous dataset are shown with an overlay of the superset of all annotated interactions below the diagonal. Gene annotations are shown below the contact maps and ChIP-seq tracks (Supplementary Table 1) are shown below the contact maps, while signal intensity scales are shown next to the maps.

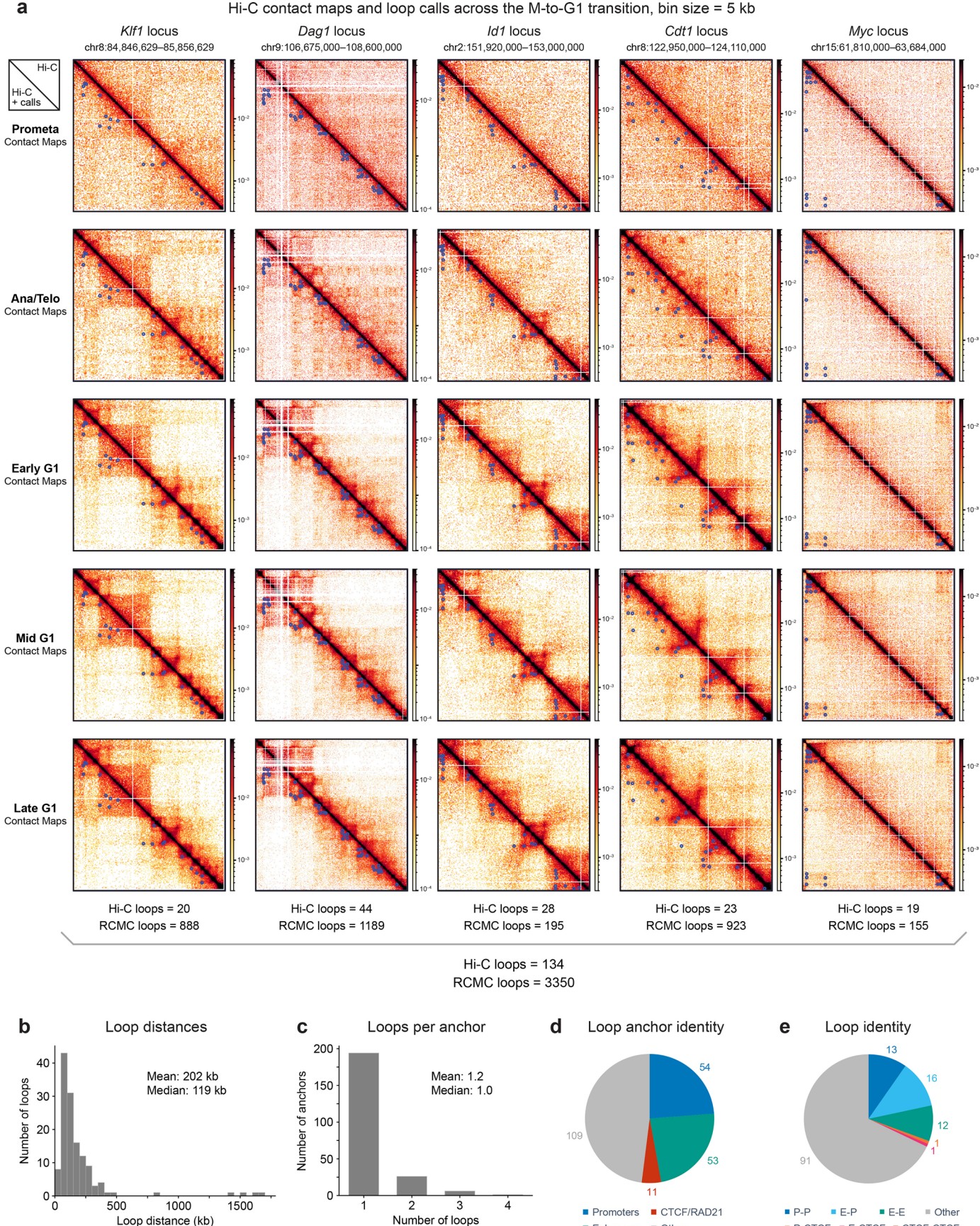

**Extended Data Fig. 8 | See next page for caption.**

**Extended Data Fig. 8 | Hi-C detects far fewer loops within the five RCMC regions.** (**a**) Array of contact maps showing previously published Hi-C data[18] and loop calls (below the diagonal) for all M-to-G1 timepoints and capture loci. The superset of M-to-G1 loop calls was generated by merging calls within 10 kb of each other into a single averaged point coordinate. All maps are shown at 5 kb bin size, and the number of loops called in Hi-C and RCMC for each locus is noted below. (**b-c**) Histograms of (**b**) loop interaction distances and (**c**) the number of loops formed by each called anchor. (**d**) Venn diagram of annotated loop anchors by their genomic identity, determined by chromatin features within 1 kb of anchor sites. Promoters were identified as annotated transcription start sites ±2 kb, enhancers as non-promoter regions with overlapping H3K4me1 and H3K27ac ChIP-seq peaks, and CTCF/RAD21 as non-promoter and non-enhancer sites with overlapping CTCF and RAD21 ChIP-seq peaks. Anchors with multiple overlapping genomic features were hierarchically classified into a single classification, with promoters taking precedence, then enhancers, and finally CTCF/RAD21. Anchors designated as "other" do not overlap promoters, enhancers, nor CTCF/RAD21. (**e**) Venn diagram of annotated loops by the genomic identity assigned in (**d**), with P designating promoters, E designating enhancers, and CTCF designating CTCF/RAD21.

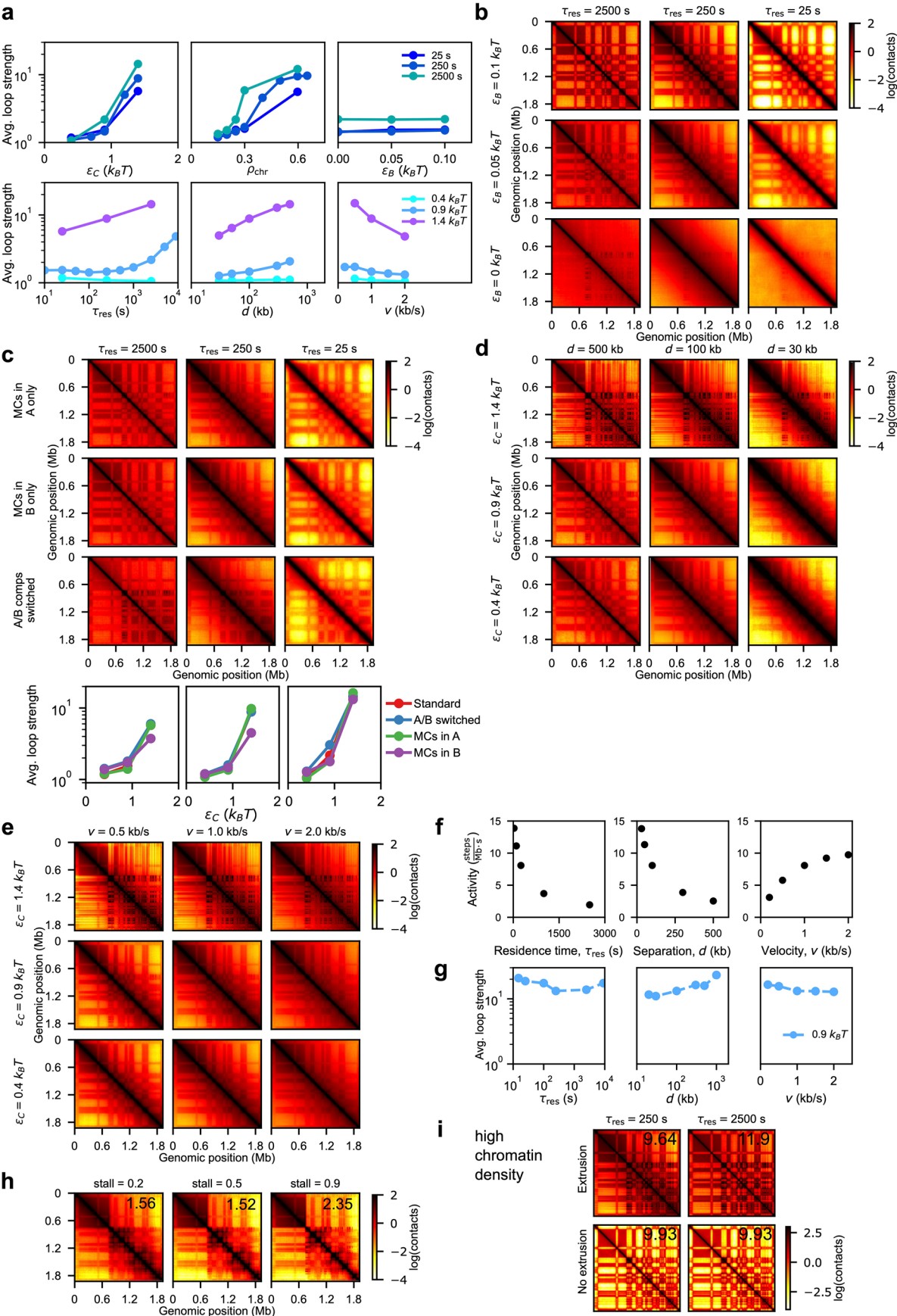

**Extended Data Fig. 9 | See next page for caption.**

**Extended Data Fig. 9 | Simulation parameter sweeps of A/B compartment strength, extruder linear density, extruder velocity, and other variables.**
(**a**) Plots of average loop strengths for different simulation parameters. Top row, from left to right: loop strength versus microcompartment affinity, $\epsilon_C$; chromatin density, $\rho_{chr}$; and A/B compartment affinity, $\epsilon_B$, for different loop extruder residence times, $\tau_{res}$ (different colors). Bottom row, from left to right: loop strength versus loop extruder residence times, $\tau_{res}$; mean separations, $d$, between loop extruders; and loop extrusion velocities, $v$, for different microcompartment affinities, $\epsilon_C$ (different colors). (**b**) Contact maps from steady-state simulations of the *Dag1* region for different loop extruder residence times, $\tau_{res}$ (decreasing from left to right columns) and A/B compartment affinities $\epsilon_B$ (decreasing from top to bottom rows). (**c**) *Top:* Contact maps with different compartment and microcompartment configurations. "MCs in A only" or "MCs in B only" denotes simulations in which only those microcompartments embedded within, respectively, A or B compartments were included. "A/B comps switched" denotes simulations with all microcompartments, but with A and B compartment identities inverted from standard simulations. *Bottom:* Average loop strengths for different compartment/microcompartment configurations. (**d**) Steady-state

simulations with loop extruder linear densities, $1/d$ (increasing from left to right) and microcompartment affinities, $\epsilon_C$ (decreasing from top to bottom), and (**e**) loop extrusion velocities, $v$ (increasing from left to right) and microcompartment affinities, $\epsilon_C$ (decreasing from top to bottom). Velocity is given as the speed of loop growth, that is, two times the mean translocation speed of each side of a loop extruder. (**f**) Extrusion activity, measured in total extrusion steps per Mb per second, as function of $\tau_{res}$, $d$, or $v$. Standard errors are smaller than the size of the data points. (**g**) Plots of average loop strengths for simulations with quenched static loops, where loop distributions were generated from extrusion simulations with different $\tau_{res}$, $d$, and $v$. Standard error for each data point was <3% of the reported value (less than the size of a data point). (**h**) Contact maps from simulations where microcompartment sites, in addition to self-affinity, also stall loop extrusion with a fixed probability. Numbers in the upper right corners of the maps indicate mean loop strengths. (**i**) Contact maps in simulations with high chromatin density, $\rho_{chr} = 0.65$, with and without loop extrusion at two different residence times. Numbers in upper right corners of maps indicate loop strengths.

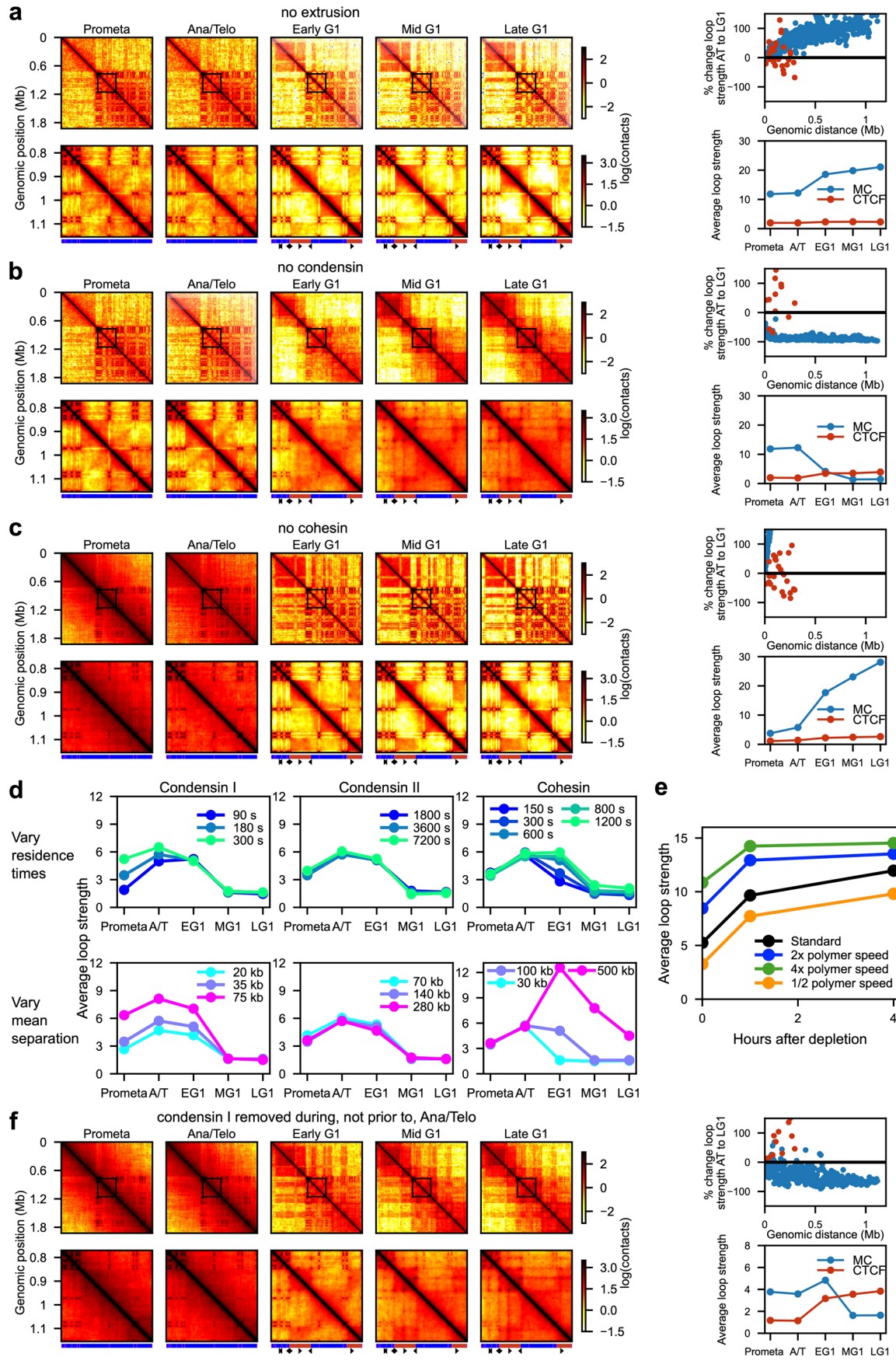

**Extended Data Fig. 10 | See next page for caption.**

**Extended Data Fig. 10 | Simulations of the mitosis-to-G1 transition without condensin and/or cohesin and with different loop extrusion parameters.** Results from simulations of the mitosis-to-G1 with (**a**) no loop extruders present, (**b**) no condensin present, but with cohesin in interphase, and (**c**) no cohesin present, but with condensins I and II in prometaphase and ana/telophase. Left panels show contact maps from various times with the top row showing the full *Dag1* region and the bottom row showing a zoomed-in view of the region marked by the box in the top row. Compartment structure and CTCF sites are indicated below. Right panels show quantification of percent change in loop strength of simulated microcompartments from ana/telophase to late G1 as a function of loop size (top) and average microcompartment and CTCF loop strengths (bottom) throughout the mitosis-to-G1 transition. (**d**) Plots of average microcompartment loop strengths throughout the mitosis-to-G1 transition

simulations for different loop extrusion parameters. Top row shows loop strengths for different residence times, $\tau_{res}$, and bottom row shows loop strengths for different mean separations, $d$. Different columns show loop strengths for alterations to different loop extruders with parameters for other loop extruders held fixed. (**e**) Average loop strengths in simulations after condensin depletion. Chromosomes are initialized with condensin prior to prometaphase, as in typical simulations. At $t = 0$ h, condensins are removed from the chromosome. Different colors indicate different relative speeds of polymer dynamics, tuned by adjusting the number of polymer timesteps between loop extrusion timesteps (see Methods). (**f**) Results from simulations of the mitosis-to-G1 with condensin I removal during ana/telophase, rather than prior to ana/telophase as in typical simulations.

# Reporting Summary

## Statistics

For all statistical analyses, confirm that the following items are present in the figure legend, table legend, main text, or Methods section.

| n/a | Confirmed | |
|---|---|---|
| ☐ | ☒ | The exact sample size (*n*) for each experimental group/condition, given as a discrete number and unit of measurement |
| ☐ | ☒ | A statement on whether measurements were taken from distinct samples or whether the same sample was measured repeatedly |
| ☒ | ☐ | The statistical test(s) used AND whether they are one- or two-sided<br>*Only common tests should be described solely by name; describe more complex techniques in the Methods section.* |
| ☒ | ☐ | A description of all covariates tested |
| ☒ | ☐ | A description of any assumptions or corrections, such as tests of normality and adjustment for multiple comparisons |
| ☐ | ☒ | A full description of the statistical parameters including central tendency (e.g. means) or other basic estimates (e.g. regression coefficient) AND variation (e.g. standard deviation) or associated estimates of uncertainty (e.g. confidence intervals) |
| ☒ | ☐ | For null hypothesis testing, the test statistic (e.g. *F*, *t*, *r*) with confidence intervals, effect sizes, degrees of freedom and *P* value noted<br>*Give P values as exact values whenever suitable.* |
| ☒ | ☐ | For Bayesian analysis, information on the choice of priors and Markov chain Monte Carlo settings |
| ☒ | ☐ | For hierarchical and complex designs, identification of the appropriate level for tests and full reporting of outcomes |
| ☒ | ☐ | Estimates of effect sizes (e.g. Cohen's *d*, Pearson's *r*), indicating how they were calculated |

*Our web collection on statistics for biologists contains articles on many of the points above.*

## Software and code

Policy information about availability of computer code

| Data collection | No software was used for data collection. RCMC analysis code is available on GitHub at https://github.com/ahansenlab/RCMC_mitosis_analysis_code and polymer simulation code is also available on GitHub at https://github.com/mirnylab/microcompartments |
|---|---|
| Data analysis | bcl2fastq v2.20.0.422, fastqc v0.11.9, bwa-mem2 v2.2.1, pairtools v0.3.0, pairix v0.3.7, cooler v0.8.11, higlass v0.8.0, cooltools v0.5.0, coolbox v0.3.3, crossmap v0.6.1, IGV v2.10.3, hicrep v1.12.2, mustache v1.2.4, bigWigToBedGraph v377, MACS2 v2.2.7.1, FIMO v5.4.1, bedtools v2.30.0, R v4.1.2, deeptools v3.5.1, Python 3.7.12, Python 3.11.5 (simulations), polychrom v0.1.0, OpenMM v8.1.0<br>RCMC analysis code is available on GitHub at https://github.com/ahansenlab/RCMC_mitosis_analysis_code and polymer simulation code is also available on GitHub at https://github.com/mirnylab/microcompartments |

For manuscripts utilizing custom algorithms or software that are central to the research but not yet described in published literature, software must be made available to editors and reviewers. We strongly encourage code deposition in a community repository (e.g. GitHub). See the Nature Portfolio guidelines for submitting code & software for further information.

## Data

Policy information about availability of data

All manuscripts must include a data availability statement. This statement should provide the following information, where applicable:

- Accession codes, unique identifiers, or web links for publicly available datasets
- A description of any restrictions on data availability
- For clinical datasets or third party data, please ensure that the statement adheres to our policy

> Sequencing data is available at NCBI Gene Expression Omnibus under accession number GSE276657 at https://www.ncbi.nlm.nih.gov/geo/query/acc.cgi?acc=GSE276657

## Research involving human participants, their data, or biological material

Policy information about studies with human participants or human data. See also policy information about sex, gender (identity/presentation), and sexual orientation and race, ethnicity and racism.

| | |
|---|---|
| Reporting on sex and gender | Not applicable. |
| Reporting on race, ethnicity, or other socially relevant groupings | Not applicable. |
| Population characteristics | Not applicable. |
| Recruitment | Not applicable. |
| Ethics oversight | Not applicable. |

Note that full information on the approval of the study protocol must also be provided in the manuscript.

# Field-specific reporting

Please select the one below that is the best fit for your research. If you are not sure, read the appropriate sections before making your selection.

☒ Life sciences    ☐ Behavioural & social sciences    ☐ Ecological, evolutionary & environmental sciences

For a reference copy of the document with all sections, see nature.com/documents/nr-reporting-summary-flat.pdf

# Life sciences study design

All studies must disclose on these points even when the disclosure is negative.

| | |
|---|---|
| Sample size | Sample sizes were determined based on the sample sizes chosen for similar experiments and analyses using other chromosome conformation capture techniques. Specifically, prior Micro-C (e.g., Hsieh et al., (2020)) and Capture (e.g., Goel et al., (2023)) studies reflected the cellular inputs required to achieve sufficient library complexity for fine-scale genome architecture mapping as well as the sequencing depth necessary to resolve structures. |
| Data exclusions | No data were excluded. |
| Replication | All RCMC maps in this study were highly reproducible. Specifically, reproducibility was determined quantitatively using HiCRep as shown in the manuscript. |
| Randomization | Not applicable. |
| Blinding | Not applicable. |

# Reporting for specific materials, systems and methods

We require information from authors about some types of materials, experimental systems and methods used in many studies. Here, indicate whether each material, system or method listed is relevant to your study. If you are not sure if a list item applies to your research, read the appropriate section before selecting a response.

## Materials & experimental systems

| n/a | Involved in the study |
|-----|-----------------------|
| ☒ | ☐ Antibodies |
| ☐ | ☒ Eukaryotic cell lines |
| ☒ | ☐ Palaeontology and archaeology |
| ☒ | ☐ Animals and other organisms |
| ☒ | ☐ Clinical data |
| ☒ | ☐ Dual use research of concern |
| ☒ | ☐ Plants |

## Methods

| n/a | Involved in the study |
|-----|-----------------------|
| ☒ | ☐ ChIP-seq |
| ☐ | ☒ Flow cytometry |
| ☒ | ☐ MRI-based neuroimaging |

# Eukaryotic cell lines

Policy information about cell lines and Sex and Gender in Research

| | |
|---|---|
| Cell line source(s) | The G1E-ER4 cell line was a gift from the Mitchell J. Weiss laboratory at St. Jude Children's Hospital. The SMC2-AID G1E-ER4 subline was previously generated and reported (Zhao et al., (2024)), and was provided by coauthor Haoyue Zhang. |
| Authentication | We regularly confirm that these cells can be induced to undergo terminal erythroid differentiation. All cell lines utilized in this study have previously been validated by PCR, sequencing, and western blotting, and by fluorescence microscopy where appropriate. |
| Mycoplasma contamination | G1E-ER4 cells have been tested to be negative for mycoplasma. |
| Commonly misidentified lines (See ICLAC register) | No commonly misidentified cell lines were used. |

# Plants

| | |
|---|---|
| Seed stocks | Not applicable. |
| Novel plant genotypes | Not applicable. |
| Authentication | Not applicable. |

# Flow Cytometry

## Plots

Confirm that:

☒ The axis labels state the marker and fluorochrome used (e.g. CD4-FITC).

☒ The axis scales are clearly visible. Include numbers along axes only for bottom left plot of group (a 'group' is an analysis of identical markers).

☒ All plots are contour plots with outliers or pseudocolor plots.

☒ A numerical value for number of cells or percentage (with statistics) is provided.

## Methodology

| | |
|---|---|
| Sample preparation | Please see the Methods section. |
| Instrument | Beckman Coulter MoFlo Astrios EQ sorter. |
| Software | FlowJo v.10.8.1. |
| Cell population abundance | We obtained generally obtain >95% viable cells based on FSC-A and SSC-A. |
| Gating strategy | To isolate pure cell populations in PM, AT, early G1, mid G1 and late G1, the following markers were used to isolate specific cell populations: prometaphase – high mCherry-MD, positive pMPM2, 4N DAPI, ana/telophase- low mCherry-MD, 4N DAPI, G1 populations- negative mCherry-MD, 2N DAPI. Sorted cells were aliquoted and flash frozen in liquid nitrogen. |

To isolate pure cell populations before and after SMC2 degradation, cells were subjected to flow cytometry to enrich for prometaphase-arrested samples. All samples were sorted for pMPM2+ cells; auxin-treated cells were sorted based on low mCherry signal (indicative of SMC2 degradation). Please see methods section for the antibody dilutions.

☒ Tick this box to confirm that a figure exemplifying the gating strategy is provided in the Supplementary Information.

