## [Peer Review File · Nature Structural & Molecular Biology]

Dynamics of microcompartment formation at the mitosis-to-G1 transition

Corresponding Author: Professor Anders Hansen

Version 0:

Decision Letter:

Our ref: NSMB-A51113-T

14th Jul 2025

Dear Dr. Hansen,

I am writing on Dimitris' behalf, as you know, as he's out of the office.

Thank you for submitting your revised manuscript "Dynamics of microcompartment formation at the mitosis-to-G1 transition" (NSMB-A51113-T). It has now been seen by the original referees and their comments are below. Thank you for providing responses to the remaining points from Rev#3. The reviewer has seen your responses and found them reasonable. Therefore, we'll be happy in principle to publish your manuscript in Nature Structural & Molecular Biology, pending minor revisions to satisfy the referees' final requests and to comply with our editorial and formatting guidelines.

We are now performing detailed checks on your paper and will send you a checklist detailing our editorial and formatting requirements after Dimitris' return. Please do not upload the final materials and make any revisions until you receive this additional information from us. I apologize for the delay while Dimitris is out of the office.

To facilitate our work at this stage, it is important that we have a copy of the main text as a Word file. If you could please send along a Word version of this file as soon as possible, we would greatly appreciate it; please make sure to copy the NSMB account (cc'ed above).

Thank you again for your interest in Nature Structural & Molecular Biology. Please do not hesitate to contact me if you have any questions.

Sincerely,

Melina Casadio, PhD
Locum Chief Editor, Nature Structural & Molecular Biology
ORCID ID: <https://orcid.org/0000-0003-2389-2243>

Reviewer #1 (Remarks to the Author):

I commend the authors for the vast improvement to this study.
I would like to note that my comment that the Blobel lab had observed contacts between CREs already at anaphase was not meant to diminish the novelty of the observations in this study. That being said, I think the re-writing of the paper to focus on the novelty of observing contacts in prophase is an important point especially considering—as the authors point out—that so many have published on the absence of 3D structure early in mitosis.
I really appreciated the TSA-A485 experiments and believe they could be included in the paper even with the caveats mentioned by the authors. But I also respect the author's decision not to do it.
I recommend the study for publication.

Reviewer #2 (Remarks to the Author):

Overall the authors have done a fantastic job in their revision and they have answered all my comments. I only have one very minor comment that remains. It is in regarding of the edits they have made to emphasize novelty. They have written (line 93-95), "Strikingly, the maps reveal strong loops across each locus already in prometaphase and ana/telophase, overturning the paradigm that all 3D genome structure is lost in prometaphase". In the discussion they write (335-338): "Thus, our findings overturn the previously established paradigm that all patterned 3D genome structure is lost in mitosis and must be re-established de novo upon G1 entry. Instead, we show that patterned 3D structure is always present, including during mitosis, and that patterned 3D structure is dynamically changing but never absent throughout the cell cycle." I worry that this will be misinterpreted. Saying that they overturn "the paradigm that all 3D genome structure is lost in prometaphase" makes it seem that they are saying that things like TADs, loops, and A/B compartments are in fact not lost in mitosis. I get the impression that their data indeed support the notion that these features are lost in mitosis, but that microcompartments specifically are not. I think that this would be clearer and would still convey the novelty of the findings if it was re-written to something along the lines of "Microcompartments are unique amongst 3D genome features as being retained during mitosis."

Reviewer #3 (Remarks to the Author):

The authors have thoroughly revised their manuscript, incorporating numerous new experiments, analyses, and simulations that significantly enhance the quality of the paper and its key messages. The rebuttal letter is detailed and has addressed—or at least satisfactorily discussed—my previous concerns.

Regarding the discussion on the role of loop extrusion activity (specifically the new Extended Sup Fig. 20g with frozen loop extruders), I would like to better understand how the simulations were performed:

- (1) Did the authors use a single configuration of frozen loops across all 10 simulated trajectories, or
- (2) Did they generate a different random sample of frozen loops for each trajectory and then average over these samples (which would require a larger number of independent trajectories to ensure appropriate statistical sampling of extruder positions)?

I suspect that the first approach was used, which could indeed lead to stronger apparent looping effects. However, this may not be the case with the second approach—the one I had in mind (see Polovnikov et al., PRX)—where loop statistics are better sampled. Could the authors clarify this point?

Only by comparing the results from Option 2 with simulations using active loop extruders can one truly disentangle the effects of loop activity from those due to loop statistics.

Version 1:

Decision Letter:

8th Sep 2025

Dear Professor Hansen,

We are now happy to accept your revised paper "Dynamics of microcompartment formation at the mitosis-to-G1 transition" for publication as an Article in Nature Structural & Molecular Biology.

To assist our authors in disseminating their research to the broader community, our SharedIt initiative provides all co-authors with the ability to generate a unique shareable link that will allow anyone (with or without a subscription) to read the

published article. Recipients of the link with a subscription will also be able to download and print the PDF.

Your paper will be published online soon after we receive proof corrections and will appear in print in the next available issue. You can find out your date of online publication by contacting the production team shortly after sending your proof corrections.

Authors may need to take specific actions to achieve compliance with funder and institutional open access mandates. If your research is supported by a funder that requires immediate open access (e.g. according to <https://www.springernature.com/gp/open-science/plan-s-compliance> Plan S principles or the <https://www.springernature.com/gp/open-science/us-federal-agency-compliance> NIH public access policy) then you should select the gold OA route, and we will direct you to the compliant route where possible. Because authors warrant under our subscription licensing terms that they haven't committed to licensing any version of their article under a licence inconsistent with the terms of our agreement – including the applicable embargo period – publication under the subscription model isn't suitable for authors whose funders require no embargo.

Sincerely,

Dimitris Typas
Senior Editor
Nature Structural & Molecular Biology
ORCID: 0000-0002-8737-1319

Response to Reviewers

Reviewer #1:

Remarks to the Author:

I commend the authors for the vast improvement to this study.

I would like to note that my comment that the Blobel lab had observed contacts between CREs already at anaphase was not meant to diminish the novelty of the observations in this study. That being said, I think the re-writing of the paper to focus on the novelty of observing contacts in prophase is an important point especially considering—as the authors point out—that so many have published on the absence of 3D structure early in mitosis.

I really appreciated the TSA-A485 experiments and believe they could be included in the paper even with the caveats mentioned by the authors. But I also respect the author's decision not to do it.

I recommend the study for publication.

Response: We thank the reviewer for the kind assessment.

Reviewer #2:

Remarks to the Author:

Overall the authors have done a fantastic job in their revision and they have answered all my comments. I only have one very minor comment that remains. It is in regarding of the edits they have made to emphasize novelty. They have written (line 93-95), “Strikingly, the maps reveal strong loops across each locus already in prometaphase and ana/telophase, overturning the paradigm that all 3D genome structure is lost in prometaphase”. In the discussion they write (335-338): “Thus, our findings overturn the previously established paradigm that all patterned 3D genome structure is lost in mitosis and must be re-established de novo upon G1 entry. Instead, we show that patterned 3D structure is always present, including during mitosis, and that patterned 3D structure is dynamically changing but never absent throughout the cell cycle.” I worry that this will be misinterpreted. Saying that they overturn “the paradigm that all 3D genome structure is lost in prometaphase” makes it seem that they are saying that things like TADs, loops, and A/B compartments are in fact not lost in mitosis. I get the impression that their data indeed support the notion that these features are lost in mitosis, but that microcompartments specifically are not. I think that this would be clearer and would still convey the novelty of the findings if it was re-written to something along the lines of “Microcompartments are unique amongst 3D genome features as being retained during mitosis.”

Response: We thank the reviewer for the assessment and suggested clarification. Indeed, as the reviewer points out, TADs and A/B-compartments are also fully absent in prometaphase in RCMC and there is full correspondence between Hi-C and RCMC there. We have re-written both line 93-95 and 335-338 according to the reviewer's suggestions to hopefully avoid any confusion.

Response to Reviewer #3

The authors have thoroughly revised their manuscript, incorporating numerous new experiments, analyses, and simulations that significantly enhance the quality of the paper and its key messages. The rebuttal letter is detailed and has addressed—or at least satisfactorily discussed—my previous concerns.

Regarding the discussion on the role of loop extrusion activity (specifically the new Extended Sup Fig. 20g with frozen loop extruders), I would like to better understand how the simulations were performed: (1) Did the authors use a single configuration of frozen loops across all 10 simulated trajectories, or (2) Did they generate a different random sample of frozen loops for each trajectory and then average over these samples (which would require a larger number of independent trajectories to ensure appropriate statistical sampling of extruder positions)?

I suspect that the first approach was used, which could indeed lead to stronger apparent looping effects. However, this may not be the case with the second approach—the one I had in mind (see Polovnikov et al., PRX)—where loop statistics are better sampled. Could the authors clarify this point?

Only by comparing the results from Option 2 with simulations using active loop extruders can one truly disentangle the effects of loop activity from those due to loop statistics.

Response: We agree with Reviewer 3 that the first option – 10 polymer simulations of a single loop configuration – would not be a rigorous exploration of the importance of loop extrusion *activity* as compared to potential structural effects from “frozen” (quenched) loops. For this reason, we followed a protocol that generated many different loop configurations for the simulated *Dag1* locus, which appropriately statistically samples extruder/loop positions, as suggested in option 2.

Specifically, we performed ≥ 10 simulations for each set of parameters. For each of these simulations we generated a unique loop configuration. Importantly, each simulated polymer contained 8 copies of the simulated *Dag1* locus. Therefore, the statistics in Extended Data Figure 9g (formerly Extended Data Figure 20g) were computed from a minimum of 80 different loop configurations within the *Dag1* locus. The resulting standard error for each data point was $< 3\%$ of the reported value (i.e., less than the size of a data point in Extended Figure 9g); we remark on this point in the revised version of the legend to Extended Figure 9g.

For a visual comparison, in **Response Figure 1**, we compare the contact map resulting from one simulation (i.e., averaging 8 loop configurations within *Dag1*) to that resulting from ten simulations (80 loop configurations). With 1 simulation, contacts resulting from individual quenched loops are clearly visible and similar contact frequency to microcompartmental contacts. With 10 simulations, many contacts quenched loops are washed out; while many others remain visible, they indicate contact frequencies far below the frequency of microcompartmental contacts (note that the contact maps are displayed with colors representing $\log(\text{contacts})$). We note that even with a single simulation, microcompartment contact frequencies are similar to those computed by averaging over ten simulations.

Response Figure 1. Contact maps for quenched loop simulations, computed from different numbers of unique loop configurations. *Top left:* Contact map for the simulated *Dag1* region computed from a single polymer simulation (8 total loop configurations within *Dag1*). *Top right:* Contact map computed from ten simulations (80 total loop configurations). Both maps were computed from simulations performed with the default parameters given in **Supplementary Table 2** (with the caveat that active loop extrusion was frozen prior to polymer equilibration and subsequent recording of data). *Bottom left:* Contact map for a 0.385 Mb snippet computed from a single simulation. *Bottom right:* Contact map for a 0.385 Mb snippet computed from ten simulations.

To address the Reviewer's question, we have revised the relevant sentences from the Methods section to the following (revised text italicized):

For simulations with quenched (inactive) loops (as in **Extended Data Figure 9g**), 1D active extrusion simulations were performed for 10^6 extrusion steps to allow the distribution of loops to equilibrate. Loop extruders were then frozen in place on the polymer chain and maintained their polymer loops throughout the remainder of the simulation. *This protocol was performed for each of at least 10 simulations, generating a unique loop configuration for each polymer simulation; given that each polymer contained 8 copies of the simulated *Dag1* locus, we simulated ≥ 80 static loop configurations within *Dag1* for each set of parameters.*